# Scaling Data-Constrained Language Models

**Niklas Muennighoff** [1]     **Alexander M. Rush** [1]     **Boaz Barak** [2]     **Teven Le Scao** [1]

**Aleksandra Piktus** [1]   **Nouamane Tazi** [1]   **Sampo Pyysalo** [3]   **Thomas Wolf** [1]   **Colin Raffel** [1]

[1] Hugging Face     [2] Harvard University     [3] University of Turku

n.muennighoff@gmail.com

## Abstract

The current trend of scaling language models involves increasing both parameter count and training dataset size. Extrapolating this trend suggests that training dataset size may soon be limited by the amount of text data available on the internet. Motivated by this limit, we investigate scaling language models in data-constrained regimes. Specifically, we run a large set of experiments varying the extent of data repetition and compute budget, ranging up to 900 billion training tokens and 9 billion parameter models. We find that with constrained data for a fixed compute budget, training with up to 4 epochs of repeated data yields negligible changes to loss compared to having unique data. However, with more repetition, the value of adding compute eventually decays to zero. We propose and empirically validate a scaling law for compute optimality that accounts for the decreasing value of repeated tokens and excess parameters. Finally, we experiment with approaches mitigating data scarcity, including augmenting the training dataset with code data or removing commonly used filters. Models and datasets from our 400 training runs are freely available at https://github.com/huggingface/datablations.

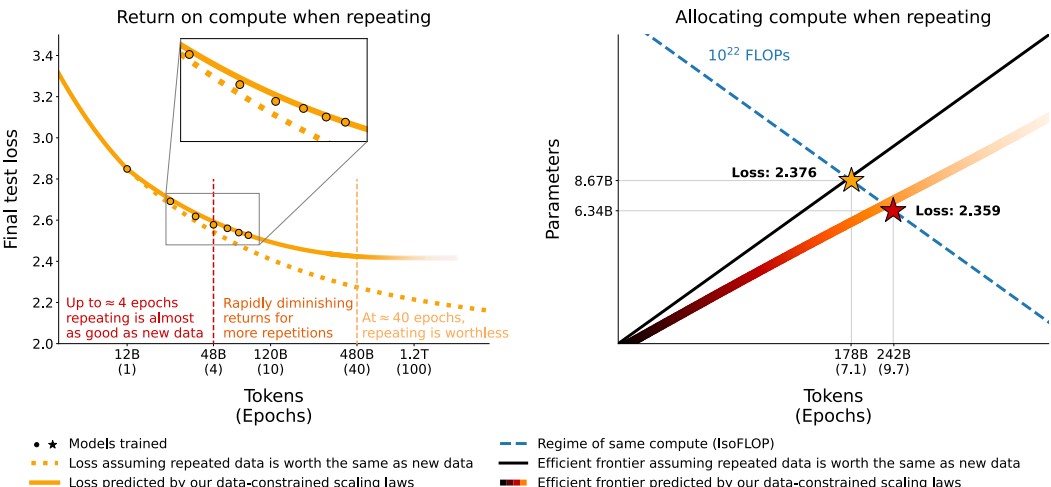

Figure 1: ***Return*** and ***Allocation*** **when repeating data.** *(Left):* Loss of LLMs (4.2B parameters) scaled on repeated data decays predictably (§6). *(Right):* To maximize performance when repeating, our data-constrained scaling laws and empirical data suggest training smaller models for more epochs in contrast to what assuming Chinchilla scaling laws [42] hold for repeated data would predict (§5).

37th Conference on Neural Information Processing Systems (NeurIPS 2023).

# 1 Introduction

Recent work on compute-optimal language models [42] shows that many previously trained large language models (LLMs, which we define as having more than one billion parameters) could have attained better performance for a given compute budget by training a smaller model on more data. Notably, the 70-billion parameter Chinchilla model [42] outperforms the 280-billion parameter Gopher model [89] while using a similar compute budget by being trained on four times more data. Extrapolating these laws for compute allocation (hereafter "Chinchilla scaling laws") to a 530 billion parameter model, such as the under-trained MT-NLG model [99], would require training on a massive 11 trillion tokens, corresponding to more than 30 terabytes of text data. For most languages, available data is several orders of magnitude smaller, meaning that LLMs in those languages are already data-constrained. Villalobos et al. [112] estimate that even high-quality English language data will be exhausted by the year 2024 given the Chinchilla scaling laws and the trend of training ever-larger models. This motivates the question [112, 81]: what should we do when we run out of data?

In this work we investigate scaling large language models in a data-constrained regime, and whether training an LLM with multiple epochs of repeated data impacts scaling. Using multiple epochs is, of course, standard in machine learning generally; however, most prior large language models have been trained for a single epoch [51, 15] and some work explicitly advocates against reusing data [40]. An exception is the recent Galactica models [108] that were trained for 4.25 epochs and exhibit continually decreasing validation loss and improving downstream performance throughout training. However, the experiments of Galactica do not compare this setup to an alternative non-data-constrained model trained for one epoch on unique data. Without this comparison, it is difficult to quantify the trade-off between additional compute versus additional data collection.

Our main focus is to quantify the impact of multiple epochs in LLM training such that practitioners can decide how to allocate compute when scaling models. Toward this end, we assembled a battery of empirical training runs of varying data and compute constraints. Specifically, we train more than 400 models ranging from 10 million to 9 billion parameters for up to 1500 epochs and record final test loss. We use these results to fit a new *data-constrained scaling law* that generalizes the Chinchilla scaling law [42] to the repeated data regime and yields a better prediction of loss in this setting. Figure 1 summarizes our main results targeting the value of repeated data (*Return*) and optimal allocation of resources in that regime (*Allocation*). We find that, while models trained for a single epoch consistently have the best validation loss per compute, differences tend to be *insignificant* among models trained for up to 4 epochs and do not lead to differences in downstream task performance. Additional epochs continue to be beneficial, but returns eventually diminish to zero. We find that, in the data-constrained regime, allocating new compute to both more parameters and epochs is necessary, and that epochs should be scaled slightly faster. These findings suggest a simple way to continue scaling total training compute budgets further ahead in the future than the previously anticipated limits.

Finally, given the challenges imposed by data constraints, we consider methods complementary to repeating for improving downstream accuracy without adding new natural language data. Experiments consider incorporating code tokens and relaxing data filtering. For code, English LLMs, such as PaLM [19] or Gopher [89], are trained on a small amount of code data alongside natural language data, though no benchmarking was reported to justify that decision. We investigate training LLMs on a mix of language data and Python data at 10 different mixing rates and find that mixing in code is able to provide a $2\times$ increase in effective tokens even when evaluating only natural language tasks. For filtering, we revisit perplexity and deduplication filtering strategies on both noisy and clean datasets and find that data filtering is primarily effective for noisy datasets.

# 2 Background

Predicting the scaling behavior of large models is critical when deciding on training resources. Specifically, two questions are of interest: *(Allocation)* What is the optimal balance of resources? *(Return)* What is the expected value of additional resources? For scaling LLMs, the resource is compute (measured in FLOPs), and it can be allocated to training a larger model or training for more

steps.[1] The metric used to quantify progress is the model's loss on held-out data, i.e. the ability to predict the underlying data as measured in the model's cross-entropy [2, 42]. We aim to minimize the loss ($L$) subject to a compute resource constraint ($C$) via optimal allocation to $N$ and $D$ as:

$$\underset{N,D}{\arg\min}\, L(N, D) \text{ s.t. FLOPs}(N, D) = C \tag{1}$$

Currently, there are established best practices for scaling LLMs. *Return* follows a power-law: loss scales as a power-law with the amount of compute used for training [39, 46, 6, 35, 7, 41]. *Allocation* is balanced: resources are divided roughly equally between scaling of parameters and data [42]. These scaling laws were established empirically by training LLMs and carefully extrapolating behavior.

Chinchilla [42] uses three methods for making scaling predictions:

- (*Fixed Parameters*) Train with a fixed model size but on varying amounts of data.
- (*Fixed FLOPs*) Train with fixed computation while parameters and training tokens vary.
- (*Parametric Fit*) Derive and fit a formula for the loss.

For the parametric fit, the loss ($L$) is a function of parameters ($N$) and training tokens ($D$):

$$L(N, D) = \frac{A}{N^\alpha} + \frac{B}{D^\beta} + E \tag{2}$$

Where $\{A, \alpha, B, \beta, E\}$ are learned variables fit using the training runs from the first two approaches [42]. Using these learned variables, they propose calculating the optimal allocation of compute ($C$) to $N$ and $D$ as follows:

$$N_{opt}(C) = G(C/6)^a \quad D_{opt}(C) = G^{-1}(C/6)^b$$
$$\text{where} \quad G = \left(\frac{\alpha A}{\beta B}\right)^{\frac{1}{\alpha+\beta}} \quad a = \frac{\beta}{\alpha + \beta} \quad b = \frac{\alpha}{\alpha + \beta} \tag{3}$$

These methods lead to the conclusion that $\alpha \approx \beta$ and hence $N$ and $D$ should be scaled proportionally for compute-optimal training. As loss can be an imperfect proxy for performance on natural language tasks [123, 97, 105], they also validate their conclusions on various downstream tasks.

## 3 Method: Data-Constrained Scaling Laws

We are interested in scaling behavior in the data-constrained regime. Specifically, given a limited amount of unique data, what is the best *Allocation* of and *Return* for computational resources. Prior work [46, 42] assumes that the necessary data to support scaling is unlimited. Our aim is therefore to introduce a modified version of Equation 2 that accounts for data constraints and fit the terms in the modified scaling law to data from a large body of experiments.

The primary method we consider is *repeating* data, i.e. allocating FLOPs to multiple epochs on the same data. Given a budget of unique data $D_C$, we split the Chinchilla total data term $D$ into two parts: the number of unique tokens used, $U_D$, and the number of repetitions, $R_D$ (i.e. epochs - 1). Given total training tokens $D$ and data budget $D_C$ these terms are simply computed as $U_D = \min\{D_C, D\}$ and $R_D = (D/U_D) - 1$. When training for a single epoch like done in prior scaling studies, $R_D = 0$. We are thus interested in minimizing Equation 1 with the additional constraint of a data budget $D_C$:

$$\underset{N,D}{\arg\min}\, L(N, D) \text{ s.t. FLOPs}(N, D) = C, U_D \leq D_C \tag{4}$$

Symmetrically, for mathematical convenience, we split the parameter term $N$ into two parts: the base number of parameters needed to optimally fit the unique tokens $U_N$, and the number of times

---

[1]In this work we use [46]'s approximation for the compute cost: FLOPs($N, D$) $\approx 6ND$, where N denotes the number of model parameters and D denotes the number of tokens processed.

to "repeat" this initial allocation, $R_N$. We compute $U_N$ by first rearranging Equation 3 to find the optimal compute budget for the unique tokens used ($U_D$). We input this value into the $N_{opt}$ formula of Equation 3 to get $U_N = \min\{N_{opt}, N\}$. $U_N$ thus corresponds to the compute-optimal number of parameters for $U_D$ or less if $N < N_{opt}$. Once we have $U_N$, we compute the repeat value as $R_N = (N/U_N) - 1$.

To empirically explore the scaling behavior in a data-limited setting we train LLMs under these constraints. We consider three different experimental protocols in this work:

- (*Fixed Unique Data*) In §5 we fix the data constraint $D_C$ and train models varying epochs and parameters. These experiments target *Allocation*, specifically tradeoff of $D$ and $N$.

- (*Fixed FLOPs*) In §6 we fix the computation available and vary $D_C$ (and thus also $U_D$ and $U_N$). These experiments target *Return*, i.e. how well does repeating scale compared to having more unique data.

- (*Parametric Fit*) We fit a formula introduced in §3.1 on all our training runs and evaluate its predictive capability throughout §5 and §6.

Before discussing experimental results we describe the parametric assumptions.

## 3.1 Parametric Fit

To extrapolate scaling curves, it is necessary to incorporate repetition into the Chinchilla formula (Equation 2). We generalize Equation 2 by replacing $D$ and $N$ with terms corresponding to the *effective data* ($D'$) and *effective model parameters* ($N'$).

$$L(N, D) = \frac{A}{N'^{\alpha}} + \frac{B}{D'^{\beta}} + E$$

Intuitively, $D'$ should be smaller or equal to $D$ where $D$ is the total number of processed tokens since repeated tokens provide less useful information to the model than new ones. We use an *exponential decay* formulation, where the value of a data token processed loses roughly $(1 - 1/R_D^*)$ fraction of its value per repetition, where $R_D^*$ is a learned constant. After some derivations and approximations (see Appendix A), this boils down to

$$D' = U_D + U_D R_D^* (1 - e^{\frac{-R_D}{R_D^*}}) . \tag{5}$$

Note that for $R_D = 0$ (no repetitions), $D' = U_D = D$. For $R_D \ll R_D^*$, $e^{-R_D/R_D^*} \approx 1 - \frac{R_D}{R_D^*}$ and so

$$D' \approx U_D + U_D R_D^* (1 - 1 + R_D/R_D^*) = U_D(1 + R_D) = D$$

and hence in this case, repeated data is worth almost the same as fresh data. (This is also consistent with the predictions of the "deep bootstrap" framework [76].) As $R_D$ grows, the value of repeated tokens tends to zero, and the effective data $D'$ becomes much smaller than $D$. The formula implies that no matter how many times we repeat the data, we will not get a better loss than could be obtained with a single epoch on $U_D + U_D R_D^*$ fresh tokens.

Just as processing repeated tokens yields a diminishing return, both intuitively and empirically, models with sizes that vastly outstrip the available data also offer diminishing returns per parameter. Hence we use a symmetric formula for the number of effective parameters, where again $R_N^*$ is learned,

$$N' = U_N + U_N R_N^* (1 - e^{\frac{-R_N}{R_N^*}}) . \tag{6}$$

The learned constants $R_D^*$, $R_N^*$ roughly correspond to the "half-life" of repeated data and excess parameters. For example, at $R_D = R_D^*$, the number of effective tokens $D'$ is $U_D + U_D R_D(1 - e^{-1})$ which means that the $U_D R_D$ repeated tokens are worth on average $1 - 1/e$ fraction of fresh ones.

Using a methodology similar to [42], $R_N^*$ and $R_D^*$ can be fit on empirical measurements, which yields data-driven estimates. See Appendix A for more details on the derivations and the fitting procedure.

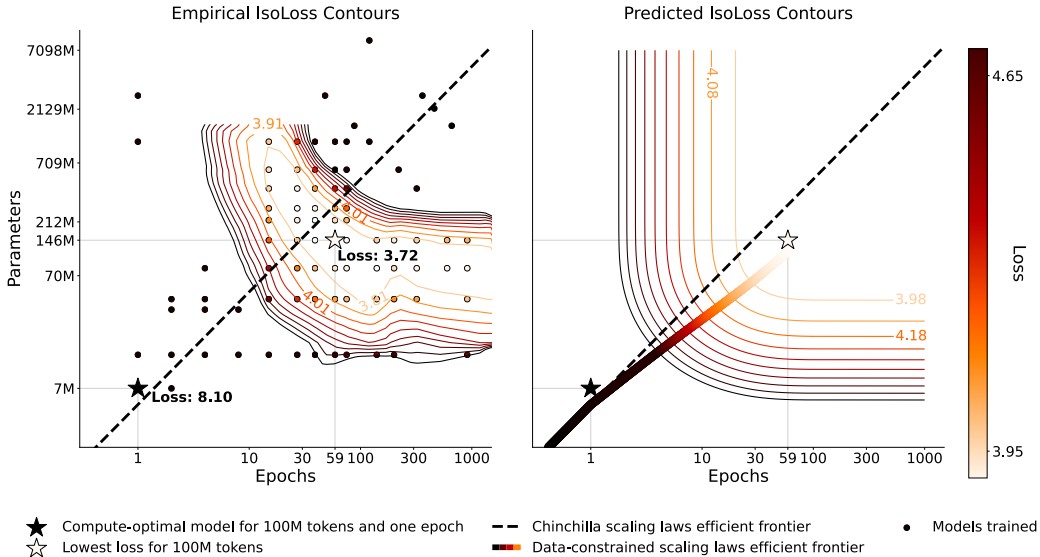

Figure 3: **IsoLoss contours for 100 million unique tokens.** *(Left):* 93 models trained with varying parameters and epochs on a fixed dataset. Contours show an interpolation of results with the same final test loss. *(Right):* Comparison with the loss predictions from our proposed scaling laws for the same budget of 100 million unique tokens and the predicted efficient frontier. The diminishing returns from training on repeated data can be seen in the increase in distance of the contour curves.

## 4   Experimental Setup

For all experiments, we train transformer language models with the GPT-2 architecture and tokenizer [88]. Models have up to 8.7 billion parameters and are trained for up to 900 billion total tokens. Following [42] we use cosine learning rate schedules that decay $10\times$ over the course of training for each model (different schedules led to different estimates in [46]). Unlike [46], we do not use early stopping to also explore the extent of overfitting when repeating. Other hyperparameters are based on prior work [89, 42] and detailed in Appendix S. Models are trained on subsets of C4 [90]. The data constraints are carefully defined to ensure maximal overlap as shown in Figure 2. Unlike [40], we always repeat the entire available data rather than subsets of it. Data is shuffled after

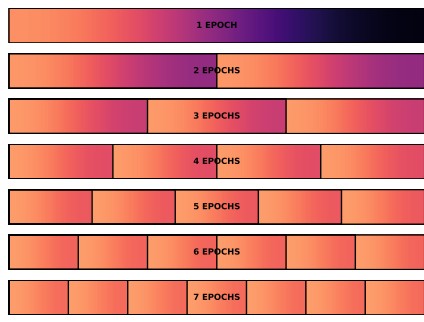

Figure 2: **Dataset setup.** We ensure that runs using less data (more epochs) always use a subset of the data used in runs with more data (fewer epochs).

each epoch. As repeating data can result in extreme overfitting (see Appendix H), we report loss on a held-out test set unless otherwise specified (see Appendix K). This contrasts training loss used in [42], but should not alter our findings as the held-out data stems from the same underlying dataset.

## 5   Results: Resource Allocation for Data-Constrained Scaling

Our first experimental setting considers scaling in a setting where all models have the same data constraint. For these experiments, the unique training data budget $D_C$ is fixed at either 100M, 400M or 1.5B tokens. For each data budget, we train a set of language models with increasing amounts of compute that is allocated to either more parameters or more epochs on the unique training data.

Figure 3 (left) shows the main results for scaling with 100M unique tokens[2] (see Appendix C for 400M and 1.5B tokens). For 100M tokens, the corresponding one-epoch compute-optimal model

---

[2]Although small, for example, this is the order of magnitude of a realistic data constraint reflecting data available after filtering the OSCAR dataset [84] for Basque, Punjabi, or Slovenian.

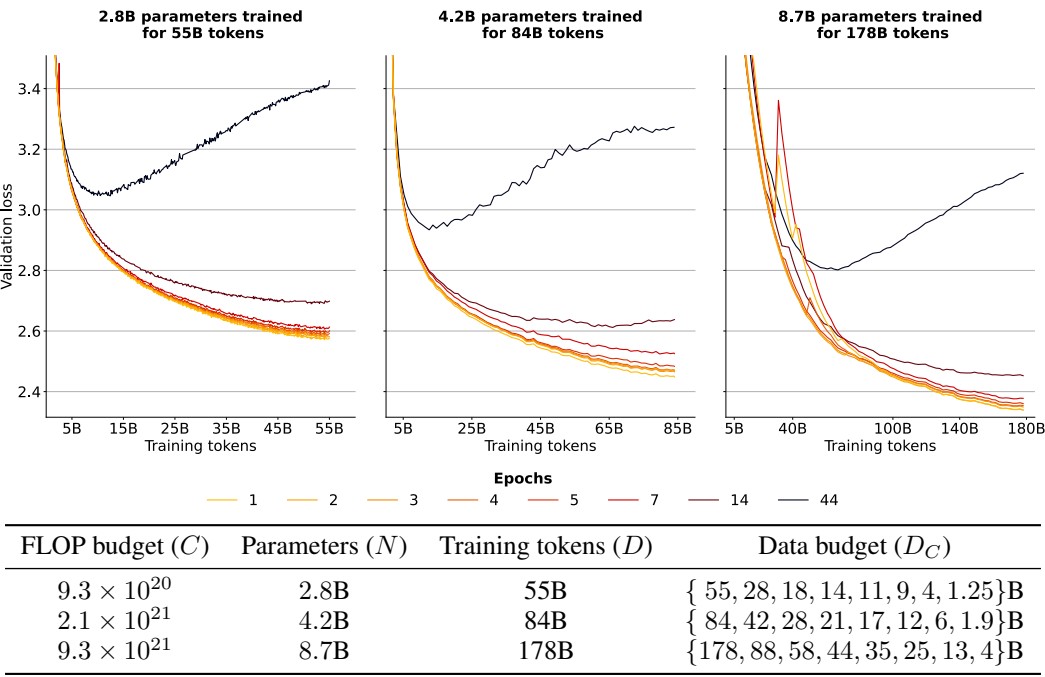

| FLOP budget ($C$) | Parameters ($N$) | Training tokens ($D$) | Data budget ($D_C$) |
|---|---|---|---|
| $9.3 \times 10^{20}$ | 2.8B | 55B | $\{\,55, 28, 18, 14, 11, 9, 4, 1.25\}$B |
| $2.1 \times 10^{21}$ | 4.2B | 84B | $\{\,84, 42, 28, 21, 17, 12, 6, 1.9\}$B |
| $9.3 \times 10^{21}$ | 8.7B | 178B | $\{178, 88, 58, 44, 35, 25, 13, 4\}$B |

Figure 4: **Validation Loss for Different Data Constraints (IsoFLOP).** Each curve represents the same number of FLOPs spent on an equal size model. Colors represent different numbers of epochs due to repeating because of data constraints. Parameters and training tokens are set to match the single-epoch compute-optimal configurations for the given FLOPs. Models trained on data that is repeated for multiple epochs have consistently worse loss and diverge if too many epochs are used.

according to scaling laws from [42] has $U_N$ of approximately 7M parameters (see Appendix B for the scaling coefficients we use). Results show that more than a 50% reduction in loss can be attained by training for several epochs ($R_D > 0$) and increasing model size beyond what would be compute-optimal for 100M tokens ($R_N > 0$). We find the best loss to be at around 20-60× more parameters and epochs, which corresponds to spending around 7000× more FLOPs. These results suggest that one-epoch models significantly under-utilize their training data and more signal can be extracted by repeating data and adding parameters at the cost of sub-optimal compute utilization.

Figure 3 (right) shows the predicted contours created by fitting our data-constrained scaling laws on 182 training runs. In the single-epoch case ($R_D = 0$) with near compute-optimal parameters ($R_N = 0$) our scaling equation (§3.1) reduces to the Chinchilla equation. In this case, both formulas predict the optimal allocation of compute to parameters and data to be the same, resulting in overlapping efficient frontiers. As data is repeated for more than a single epoch, our fit predicts that excess parameters decay faster in value than repeated data ($R_N^* < R_D^*$). As a result, the data-constrained efficient frontier suggests allocating most additional compute to more epochs rather than more parameters. This contrasts the Chinchilla scaling laws [42], which suggest equally scaling both. However, note that they do not repeat the entire training data and their parametric fit explicitly relies on the assumption that models are trained for a single epoch only. Thus, there is no guarantee that their scaling predictions hold for repeated data.

For all three data budgets, our results suggest that *Allocation* is optimized by scaling epochs faster than additional parameters. We confirm this at scale by training the data-constrained compute-optimal model for $9.3 \times 10^{21}$ FLOPs and 25 billion unique tokens as suggested by our efficient frontier. Despite having 27% less parameters, this model achieves better loss than the model suggested by the Chinchilla scaling laws (Figure 1, right). Similarly, the 120 billion parameter Galactica model trained on repeated data should have been significantly smaller according to data-constrained scaling laws (Appendix G). An additional benefit of using a smaller model is cheaper inference, though adding parameters can make it easier to parallelize training across GPUs.

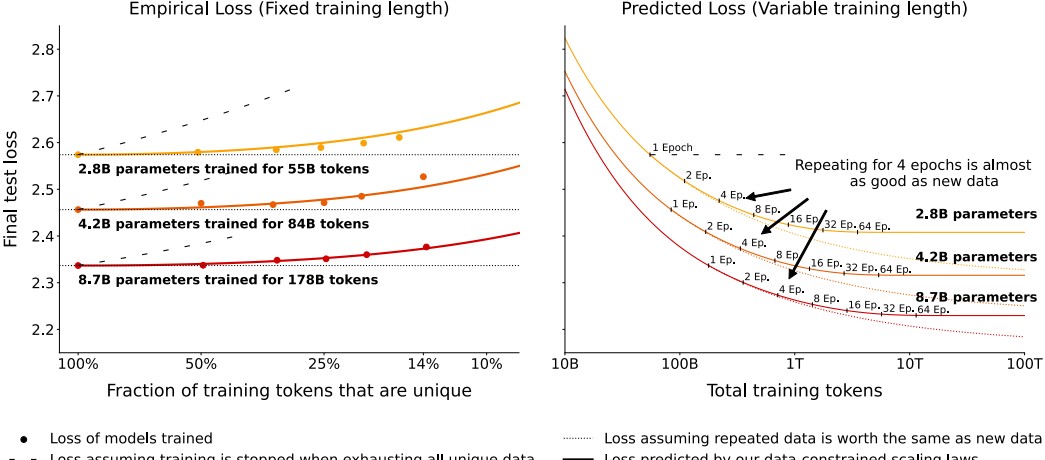

Figure 5: **Empirical and Extrapolated loss with constrained data.** *(Left):* Loss as a function of repeated tokens for three different training budgets each with fixed number of parameters. Loss curves predicted by our data-constrained scaling laws are shifted to exactly match the loss at 100% unique data. Return on FLOPs decays with repeated data in a regular pattern. *(Right):* Extrapolating from the proposed data-constrained scaling law shows that at small numbers epochs are benign, but at large number of epochs loss stops improving.

Adding parameters and epochs causes the loss to decrease and eventually increase again, suggesting that too much compute can hurt performance. Results from [46] also show that loss can increase when too many parameters are used, even with early stopping. However, we expect that appropriate regularization (such as simply removing all excess parameters as an extreme case) could prevent this behavior. Thus, our formula presented in §3 and its predicted isoLoss contours in Figure 3 do not model the possibility that excess epochs or parameters could hurt performance.

## 6   Results: Resource Return for Data-Constrained Scaling

Next, consider the question of *Return* on scaling. To quantify this value, we run experiments with three FLOP budgets across eight respective data budgets to compare return on FLOPs.

Figure 4 shows the configurations and validation curves for models trained on the same number of total tokens. Conforming to intuition and prior work on deduplication [55], repeated data is worth less, thus models trained on less unique data (and, correspondingly, more epochs) have consistently higher loss. However, the loss difference for a few epochs is negligible. For example, the $N = 8.7$ billion parameter model trained for four epochs ($D_C = 44$ billion unique tokens) finishes training with only 0.5% higher validation loss than the single-epoch model ($D_C = 178$ billion unique tokens).

In Figure 5 (left), we compare the final test loss of each model to predictions from our parametric fit. The data-constrained scaling laws can accurately measure the decay in the value of repeated data as seen by the proximity of empirical results (dots) and parametric fit (lines). We note however that it significantly underestimates the final test loss of failing models where loss increases midway through training, such as models trained for 44 epochs (not depicted).

In Figure 5 (right), we extrapolate the three budgets by further scaling compute while keeping the data constraints ($D_C$) at 55B, 84B, and 178B tokens, respectively. The parameter $R_D^*$ introduced in §3 represents roughly the "half-life" of epochs: specifically the point where repeated tokens have lost $\frac{1}{e}$ of their value. Through our fitting in Appendix A, we found $R_D^* \approx 15$, corresponding to 15 repetitions (or 16 epochs). Graphically, this can be seen by the stark diminishing returns in the proximity of the 16-epoch marker and the flattening out soon after.

Overall, the *Return* when repeating data is relatively good. Meaningful gains from repeating data can be made up to around 16 epochs ($R_D^*$) beyond which returns diminish extremely fast.

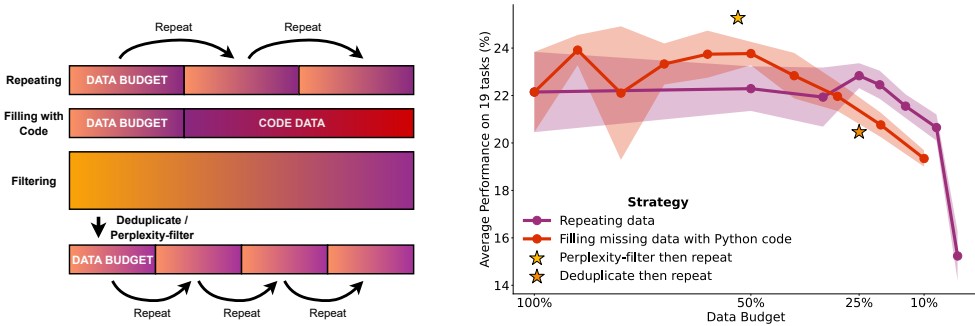

Figure 6: **Strategies for data-constrained settings and their downstream performance.** *(Left):* Schematic showing alternative data use strategies of code filling and filtering. *(Right):* $N = 4.2$ billion parameter models trained for a total of $D = 84$ billion tokens with varying budgets $D_C$. For repeating and filling with code, five models with different seeds are trained for each dot and the standard deviation is visualized as the shaded area.

# 7 Results: Complementary Strategies for Obtaining Additional Data

While repeating data is effective, it has diminishing returns. We therefore consider strategies for scaling $D$ targeting improved downstream performance as opposed to directly minimizing loss.

Figure 6 (left) illustrates the strategies: **(a) Code augmentation:** We use Python code from The Stack [49] to make up for missing natural language data. The combined dataset consisting of code and natural language samples is shuffled randomly. **(b) Adapting filtering:** We investigate the performance impact of deduplication and perplexity filtering, two common filtering steps that can severely limit available data. Removing such filtering steps can free up additional training data.

For these experiments, we set a maximum data budget ($D_C$) of 84 billion tokens. For repetition and code filling, only a subset of $D_C$ is available and the rest needs to be compensated for via repeating or adding code. For both filtering methods, we start out with approximately twice the budget (178 billion tokens), as it is easier to gather noisy data and filter it than it is to gather clean data for training. For perplexity filtering, we select the top 25% samples with the lowest perplexity according to a language model trained on Wikipedia. This results in 44 billion tokens that are repeated for close to two epochs to reach the full data budget. For deduplication filtering, all samples with a 100-char overlap are removed resulting in 21 billion tokens that are repeated for four epochs during training. See Appendix N for more details on the filtering procedures.

When comparing across data strategies, loss ceases to be a good evaluation metric as the models are trained on different data distributions. We thus evaluate models on 19 natural language tasks with zero to five in-context few-shot exemplars [15] producing 114 scores per model. As our evaluation tasks cover different metrics and random baselines, we re-scale all scores to be in the same range to better reflect performance ranges before averaging. Details on the evaluation datasets are in Appendix K.

In Figure 6 (right) we compare the downstream performance of all strategies. For repeating data, differences in downstream performance are insignificant for up to around 4 epochs (25% budget) and then start dropping, which aligns with our results on test loss in §6. Filling up to 50% of data with code (42 billion tokens) also shows no deterioration. Beyond that, performance decreases quickly on natural language tasks. However, adding more code data may benefit non-natural language tasks, which are not considered in the benchmarking. Two of the tasks benchmarked, WebNLG [17, 34], a generation task, and bAbI [122, 57], a reasoning task, see jumps in performance as soon as code is added, possibly due to code enabling models to learn long-range state-tracking capabilities beneficial for these tasks.

Of the filtering approaches, we find perplexity-filtering to be effective, while deduplication does not help. Prior work found deduplication was able to improve perplexity [55]; however, it did not evaluate on downstream tasks. Deduplication may have value not captured in our benchmark, such as reducing memorization [45, 40, 16, 10]. We also investigate filtering on a different noisier dataset in Appendix O, where we find it to be more effective. Overall, in a data-constrained regime, we recommend reserving filtering for noisy datasets and using both code augmentation and repeating to

increase data tokens. For example, first doubling the available data by adding code and then repeating the new dataset for four epochs results in $8\times$ more training tokens that are expected to be just as good as having had $8\times$ more unique data from the start.

# 8 Related Work

**Large language models** Scaling up transformer language models [111] across parameter count and training data has been shown to result in continuous performance gains [19]. Starting with the 1.4 billion parameter GPT-2 model [88], a variety of scaled-up language models have been trained, commonly referred to as large language models (LLMs). They can be grouped into dense models [15, 47, 58, 89, 20, 13, 132, 109, 103, 108, 130, 95, 56, 65] and sparse models [30, 131, 28, 135] depending on whether each forward pass makes use of all parameters. These models are generally pre-trained to predict the next token in a sequence, which makes them applicable to various language tasks directly after pre-training [15, 118, 50, 71, 102] by reformulating said NLP tasks as context continuation tasks (see [67] for an earlier proposal on this topic). We focus on the most common scenario, where a dense transformer model is trained to do next-token prediction on a large corpus and evaluated directly after pre-training using held-out loss or zero- to few-shot prompting.

**Scaling laws** Prior work has estimated an optimal allocation of compute for the training of LLMs. Kaplan et al. [46] suggested a $10\times$ increase in compute should be allocated to a $5.5\times$ increase in model size and a $1.8\times$ increase in training tokens. This first scaling law has led to the creation of very large models trained on relatively little data, such as the 530 billion parameter MT-NLG model trained on 270 billion tokens [99]. More recent work [42], however, showed that model size and training data should rather be scaled in equal proportions. These findings called for a renewed focus on the scaling of pre-training data rather than scaling model size via complex parallelization strategies [98, 91, 9, 78]. Up-sampling is often employed when pre-training data is partly limited, such as data from a high-quality domain like Wikipedia or text in a rare language for training multilingual LLMs [60, 82]. Hernandez et al. [40] study up-sampling of data subsets and find that repeating only 0.1% of training data 100 times significantly degrades performance. In contrast, our work focuses on repeating the entire pre-training corpus for multiple epochs rather than up-sampling parts of it.

**Alternative data strategies** Large pre-training datasets are commonly filtered to remove undesired samples or reduce noise [101]. Perplexity-based filtering, whereby a trained model is used to filter out samples with high perplexity, has been found beneficial to reduce noise in web-crawled datasets [121]. Mixing of data is employed for the pre-training data of multilingual LLMs, where text data from different languages is combined [23, 126, 100, 74]. However, both for code and natural language models, mixing different (programming) languages has been reported to under-perform monolingual models [80, 113]. Some work has investigated mixing code and natural language data for prediction tasks, such as summarizing code snippets [44] or predicting function names [4]. Several pre-training datasets for LLMs include low amounts of code data [31, 89, 95]. However, these past works generally do not provide any ablation on the drawbacks of including code or the benefits for natural language task performance. We perform a detailed benchmarking of mixing Python and natural language in LLM pre-training at 10 different mixing rates.

# 9 Conclusion

This work studies data-constrained scaling, focusing on the optimal use of computational resources when unique data is limited. We propose an extension to the Chinchilla scaling laws that takes into account the decay in value of repeated data, and we fit this function using a large set of controlled experiments. We find that despite recommendations of earlier work, training large language models for multiple epochs by repeating data is beneficial and that scaling laws continue to hold in the multi-epoch regime, albeit with diminishing returns. We also consider complementary approaches to continue scaling models, and find that code gives the ability to scale an additional $2\times$. We believe that our findings will enable further scaling of language models to unlock new capabilities with current data. However, our work also indicates that there are limits on the scaling horizon. In addition to collecting additional data, researchers should explore using current data in a more effective manner.

## Acknowledgments and Disclosure of Funding

This work was co-funded by the European Union under grant agreement No 101070350. The authors wish to acknowledge CSC – IT Center for Science, Finland, for generous computational resources on the LUMI supercomputer.[3] We are thankful for the immense support from teams at LUMI and AMD, especially Samuel Antao. Hugging Face provided storage and additional compute instances. This work was supported by a Simons Investigator Fellowship, NSF grant DMS-2134157, DARPA grant W911NF2010021, and DOE grant DE-SC0022199. We are grateful to Harm de Vries, Woojeong Kim, Mengzhou Xia and the EleutherAI community for exceptional feedback. We thank Loubna Ben Allal for help with the Python data and Big Code members for insightful discussions on scaling laws. We thank Thomas Wang, Helen Ngo and TurkuNLP members for support on early experiments.

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
