# Appendix

## Contents

# A Derivation of Data-Constrained Scaling Laws

Let $N$ be the number of model parameters, $D$ be the training tokens and $U$ be the "unique" training tokens i.e. the size of the dataset that is to be trained on for one or more epochs. Chinchilla [42] only deals with non-repeated tokens, thus $D = U$ and we can write their formula ("Approach 3") as:

$$L(N, U) = \frac{A}{N^\alpha} + \frac{B}{U^\beta} + E \tag{7}$$

where $E$ represents the irreducible loss. $A$, $B$, $\alpha$ and $\beta$ are learned parameters.

We now want to generalize this expression to multiple epochs where tokens are repeated. We repeat the data $R_D$ times, where $R_D = 0$ corresponds to the base case of a single epoch. We let $D'$ be the "effective data size": the number of unique data needed to get the same value as repeating $U$ unique tokens for $R_D$ repeats. Hence, if $R_D = 0$, the effective data is the same as the total data processed. Intuitively, each time a sample is repeated, it is worth less as the model has already learned some of its information. Assume that each time a model trains on a token, it learns a $1 - \delta$ fraction of the information in it for some constant $0 \leq \delta \leq 1$. (Thus, if $\delta = 0$ repeated tokens are as good as new ones, and if $\delta = 1$, repeated tokens are worth nothing.) In other words, we expect the decrease in value of each repetition to be proportional to the value of the prior repetition, which is equivalent to exponential decay. As we would like to sum up the value of all repetitions, we temporarily assume an integral number of repeats and express it as a geometric series:

$$D' = U + (1 - \delta)U + (1 - \delta)^2 U + \cdots + (1 - \delta)^{R_D} U \tag{8}$$

We know that the sum $S$ of a geometric series with a common ratio $r$ is:

$$S = \frac{a(1 - r^n)}{1 - r} \tag{9}$$

where $a$ is the first term and $n$ the number of terms in the series. As $r = (1 - \delta)$ and $a = (1 - \delta)U$:

$$D' = U + U \sum_{k=1}^{R_D} (1 - \delta)^k = U + (1 - \delta)U \frac{(1 - (1 - \delta)^{R_D})}{\delta} \tag{10}$$

Note that Equation 10 can also be used with a non-integer number of repetitions. We can directly use Equation 10 as our effective data and learn $\delta$ but for convenience and interpretability, we redefine it in terms of the number of epochs beyond which repeating does not help. Note that as more data is repeated, the right-hand side tends to $\frac{(1 - \delta)U}{\delta}$, as $\lim_{R_D \to \infty} (1 - (1 - \delta)^{R_D}) = 1$. Let $R_D^* = \frac{1 - \delta}{\delta}$, hence $D'$ "plateaus" at $U + R_D^* U$ as $R_D$ goes to infinity.

If we assume $\delta$ to be small, $1 - \delta$ tends to one and we can approximate $1/R_D^* = \frac{\delta}{1 - \delta} \approx \delta$.

Next, define $e^x$ in terms of its Taylor series expansion:

$$e^x = 1 + x + \frac{x^2}{2!} + \frac{x^3}{3!} + \cdots \approx 1 + x \tag{11}$$

If $x$ is small later terms become increasingly small, thus $e^x \approx 1 + x$. As we have assumed $\delta$ to be small, let $x = -\delta$, which yields

$$(1 + x) = (1 - \delta) \approx e^{-\delta} \approx e^{-1/R_D^*} \tag{12}$$

Now inserting $(1 - \delta)/\delta = R_D^*$ and $(1 - \delta)^{R_D} = e^{(-1/R_D^*)^{R_D}}$ into Equation 10 we get our final equation representing the *effective data*:

$$D' = U + U \cdot R_D^* \cdot (1 - e^{-R_D/R_D^*}) \tag{13}$$

where $U$ and $R_D$ are given while $R_D^*$ is a learned constant. If no repeats are done, the second part of the sum is zero and the term simplifies to the single-epoch scaling laws from Equation 7. While $R_D \ll R_D^*$, the second term is approximated as $U \cdot R_D$ and for $R_D \gg R_D^*$, it plateaus at $U \cdot R_D^*$. Hence $R_D^*$ corresponds to the number of times we can repeat tokens before seeing sharply diminishing returns.

Let us consider a concrete example to show that Equation 13 is a very good approximation of Equation 10 and make the equations more intuitive. Suppose repeated data retains 75% of its value ($\delta = 0.25$) and we train on a single token or data unit ($U = 1$) for five epochs, i.e. we repeat it four times ($R_D = 4$). In that case Equation 10 yields $D' = U + (1 - \delta)U\frac{(1-(1-\delta)^{R_D})}{\delta} = 1 + (0.75) * 4 * (1 - 0.75^4) = 3.05$. Thus despite training for 5 total units (4 of which are repetitions), we only get the value equivalent to 3.05 units. As we have defined $R_D^* = (1 - \delta)/\delta$, the corresponding $R_D^*$ value is 3. Setting $R_D^* = 3$ in Equation 13 yields $D' = U + U \cdot R_D^* \cdot (1 - e^{-R_D/R_D^*}) = 1 + 3 * (1 - e^{-4/3}) = 3.21$. Due to our approximations, the results are not the same, i.e. 3.21 is slightly higher than 3.05. However, note that the data term is additionally raised to a power of $\beta = 0.353$ (see Equation 7; Appendix B), thus the actual difference calculated as $((3.21^{0.353})/(3.05^{0.353})) - 1$ is a mere 1.8% despite this relatively large $\delta$ of 0.25. Equation 13 has the benefit that we can interpret $R_D^*$ as the number of repetitions beyond which repeating yields sharply diminishing returns and flattens out soon after. Consider $R_D = 100$ then $D' = 1 + 3 * (1 - e^{-100/3}) = 3.99$. No matter how many repeats are done the effective data will never exceed 4 i.e. it plateaus at $U + R_D^*U$ as $R_D$ tends to infinity.

Similarly, we consider repeating parameters. Symmetric to seeing the same data, excess parameters learn the same features and do not add any value in the extreme. For the Chinchilla equation (Equation 7) increasing parameters from 1 billion to 10 billion yields the same absolute decrease in loss regardless of whether the dataset is a single token or 1 billion tokens. However, intuition and our data (Appendix F) suggest that in the first case, adding parameters should not decrease loss at all, as the additional 9 billion parameters cannot possibly learn anything from the single token that the first 1 billion parameters have not already learned. Thus, to allow excess parameters to decay to adding nothing, we also replace $N$ with a symmetric version of Equation 13 yielding our final equation:

$$L(U_N, U_D, R_N, R_D) = \frac{A}{(U_N + U_N R_N^*(1 - e^{\frac{-R_N}{R_N^*}}))^\alpha} + \frac{B}{(U_D + U_D R_D^*(1 - e^{\frac{-R_D}{R_D^*}}))^\beta} + E$$
(14)

We define $U_N$, as the number of "unique" parameters that provide an optimal fit for $U_D$. Additional parameters decay with a symmetric version of the expression for repeated data. $R_N$ is the number that the "unique" parameters are repeated i.e. $R_N = \max\{(N/U_N) - 1, 0\}$. If $R_N^* = \infty$, additional parameters do not decay at all and $(U_N + U_N R_N^*(1 - e^{\frac{-R_N}{R_N^*}}))$ reduces to $N$. We compute $U_N$ from $U_D$ by setting $D_{opt} = U_D$ and rearranging Equation 3 to map from $D_{opt}$ to $N_{opt}$. $U_N$ is then $\min\{N_{opt}, N\}$. This is equivalent to the following:

$$U_N = \min\{((U_D \cdot G)^{\beta/\alpha}) \cdot G, N\} \quad \text{where} \quad G = \left(\frac{\alpha A}{\beta B}\right)^{\frac{1}{\alpha+\beta}}$$
(15)

Equation 14 is a generalization of Equation 7: It provides the same estimates for optimal model and data size in the single epoch case, but allows for decay in the value of parameters and tokens, thus generalizing to training for multiple epochs and with excess parameters. It can thus be used as a direct replacement of Equation 7. If $R_N^*$ and $R_D^*$ are unknown, one can simply set them to infinity by default, which will make Equation 14 completely equivalent to Equation 7.

To learn the parameters $R_N^*$ and $R_D^*$, we largely follow the approach from [42]. We fix $a$, $b$, $e$, $\alpha$, $\beta$ to the values learned on C4 in Appendix B and minimize:

$$\min_{R_N^*, R_D^*} \sum_{\text{Run } i} \text{Huber}_\delta \Big( \text{LSE}\big(a - \alpha \log(U_N^i + U_N^i R_N^*(1 - e^{\frac{-R_N^i}{R_N^*}})),$$
(16)
$$b - \beta \log(U_D^i + U_D^i R_D^*(1 - e^{\frac{-R_D^i}{R_D^*}})), e\big) - \log L^i \Big)$$

We use the LBFGS algorithm to find local minima of the objective above, started on a grid of initialization given by: $R_N^* \in \{0., 4., \ldots, 20.\}$ and $R_D^* \in \{0., 4., \ldots, 20.\}$. We fit on 182 samples with parameters varying from 7 million up to 9 billion and epochs ranging from 1 to 500. We removed outliers referenced in Appendix F from our fitting, as our formulas do not allow for excess parameters or excess epochs to negatively impact performance. We assume excess parameters or epochs only cause performance to plateau but never to worsen. However, it is difficult to identify all samples where excess parameters or epochs hurt, as for some data budgets we only train a single model, thus we do not know if the loss of that model is already in the range where it starts to increase again. Further, there are samples where loss initially increases and then decreases as a function of epochs (double descent, see Appendix D), which further contributes to noise in the fitting. Nevertheless, we are able to get a fairly stable fit resulting in $R_N^* = 5.309743$ and $R_D^* = 15.387756$. Since $R_D^* > R_N^*$, excess parameters decay faster. Hence, the data-constrained efficient frontiers in Figures 1,3 suggest scaling compute allocated to epochs faster than to parameters. This value of $R_D^*$ yields $\delta \approx 6 * 10^{-2}$ (0.19 for $R_N^*$), which respects the assumption that $\delta$ is small. Inserting these learned parameters and the parameters from Appendix B, and simplifying Equation 15 yields the precise formulation we use to predict loss ($L$) given unique tokens ($U_N$), parameter repetitions ($R_N$) and data repetitions ($R_D$):

$$L(U_D, R_N, R_D) = \frac{521}{(U_N + 5.3 \cdot U_N(1 - e^{\frac{-R_N}{5.3}}))^{0.35}} + \frac{1488}{(U_D + 15.4 \cdot U_D(1 - e^{\frac{-R_D}{15.4}}))^{0.35}} + 1.87$$

$$\text{where } U_N = U_D \cdot 0.051$$

$$\tag{17}$$

Table 1: **Comparison of different versions of our parametric fit.** All versions are fitted on the same 182 samples. We report the fitting loss and the $R^2$ on those samples. No decay corresponds to assuming Chinchilla holds for repeated data without modification necessary. For Equation 10, we use the same equation for $D$ and $N$ renaming the $\delta$ to $R_D^*$ and $R_N^*$.

| Parametric Fit | $R_D^*$ | $R_N^*$ | Loss | $R^2$ |
|---|---|---|---|---|
| No decay | - | - | - | 0.1430 |
| Equation 14 but only decay $N$ | - | 713.0015 | 0.0241 | 0.1671 |
| Equation 14 but only decay $D$ | 2.9157 | - | 0.0169 | 0.7395 |
| Equation 14 | 15.3878 | 5.3097 | 0.0158 | 0.7810 |
| Equation 10 for both $N$ and $D$ | 0.0104 | 0.3676 | 0.0155 | 0.8062 |
| Equation 18 for both $N$ and $D$ | 0.0105 | 0.3676 | 0.0155 | 0.8061 |

We experiment with different versions of our formula and display the learned values in Table 1. No decay or decaying only $D$ or $N$ of Equation 14 leads to worse loss and $R^2$ than Equation 14. Thus, it is important to decay both the value of excess parameters and data repetitions. We also consider an explicit exponential where $D' = \sum_{k=0}^{R_D} U * e^{-R_D^* k}$, hence from Equation 9 it follows:

$$D' = U \frac{1 - (e^{-R_D^*})^{R_D + 1}}{1 - e^{-R_D^*}} \tag{18}$$

This explicit decay, Equation 10, and Equation 14 all yield similar results with $R^2$ around 80. Equation 14 fits the data slightly worse than Equation 10, likely due to our approximations. Nevertheless, we use Equation 14 throughout as it has fewer terms, and we find it easier to interpret.

## A.1 Analytical properties of compute-optimal point

In our case, consider the setting of a fixed compute budget $C$ and a fixed budget of unique tokens $U_D$ implying a set of unique parameters $U_N$. Let $R_D$ denote the number of times we repeat data (we assume that we are in the multi-epoch regime and hence $R_D > 0$).

Write $U_D = cU_N$ (for Chinchilla $c \approx 20$). When $R_D \ll R_D^*$ and $R_N \ll R_N^*$, our scaling agrees with Chinchilla, and so the point $(U_N, U_D)$, corresponding to $R_D = R_N = 0$ is on the optimal compute curve. Increasing $R_D$ by $\epsilon$ corresponds to increasing the number of tokens by $\epsilon U_D = \epsilon cU_N$,

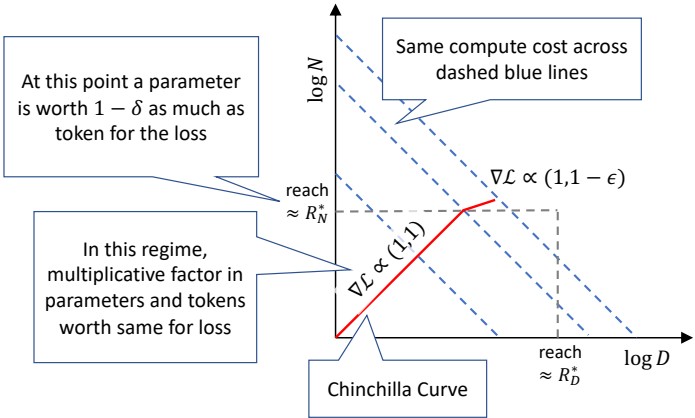

Figure 7: A cartoon of how the compute-optimal tradeoff deviates from Chinchilla as we increase the number of epochs. Initially the model size and tokens processed grow proportionally ($R_N = R_D$) but since $R_N^* < R_D^*$, at some point adding parameters offers worse returns compared to increasing the number of tokens processed, and hence we deviate from the Chinchilla curve.

while increasing $R_N$ by $\epsilon$ corresponds to increasing the number of parameters by $\epsilon U_N$. For small positive $R_D, R_N$, our curve agrees with Chinchilla and so we need to increase $R_N, R_D$ by the same amount to maintain the proportionality. Hence up to some value $r > 0$, the optimal compute curve corresponds to $R_N = R_D = r$. Our curve differs from Chinchilla when $r$ gets closer to either $R_N^*$ or $R_D^*$. At this point, we start to see sharply diminishing returns.

In our setting, $R_D^* > R_N^*$ which means that we reach the point $r \approx R_N^*$ first. At this point, each added parameter is worth less (specifically worth $e^{-r/R_N^*}$), than an added data point, despite them having equal computational cost. Hence processing more tokens will be more effective than increasing the number of parameters, and we expect the optimal compute curve to break away from proportionality. This is indeed what we see.

## B  C4 Scaling Coefficients

While Hoffmann et al. [42] have shown that the equal scaling of model parameters and training tokens holds across different training datasets, the precise ratios vary considerably across datasets and approaches. For example given the Gopher [89] compute budget of $5.76 \times 10^{23}$ FLOPs, their parametric loss function fitted on MassiveWeb predicts an optimal allocation of 40 billion parameters. Meanwhile, if the training dataset is C4 [90] their IsoFLOP approach predicts 73 billion parameters to be optimal, almost twice as much. However, for C4, which is our training dataset, they do not provide the coefficients necessary to compute loss with their parametric loss function. Based on their IsoFLOP training runs on C4, they only provide the information that for C4, compute ($C$) allocated to data ($D$) and parameters ($N$) should be scaled *exactly* equally for optimality, i.e. $a = b = 0.5$ in the relationship $N_{opt} \propto C^a$ and $D_{opt} \propto C^b$. This corresponds to $\alpha = \beta$ in the parametric loss function (Equation 2). Thus, we use this information together with the methodology and C4 data points from [42] to fit the parametric loss function. We tie the parameters $\alpha$ and $\beta$ to be equal and optimize

$$\min_{a,b,e,\alpha,\beta} \sum_{\text{Run } i} \text{Huber}_\delta \Big( \text{LSE}\big(a - \alpha \log N_i, b - \beta \log D_i, e\big) - \log L_i \Big) \tag{19}$$

where LSE is the log-sum-exp operator and $N_i$, $D_i$ and $L_i$ the model size, dataset size and loss of the $i$th run, and $\delta = 10^{-3}$. We fit on 54 samples on a grid of initialization given by: $\alpha \in \{0., 0.5, \ldots, 2.\}$, $\beta \in \{0., 0.5, \ldots, 2.\}$, $e \in \{-1, -.5, \ldots, 1.\}$, $a \in \{0, 5, \ldots, 25\}$, and $b \in \{0, 5, \ldots, 25\}$. Our fit results in $a = 6.255414$, $b = 7.3049974$, $e = 0.6254804$, $\alpha = \beta = 0.3526596$. Exponentiating $a$, $b$ and $e$ to get $A$, $B$ and $E$ and inserting all learned coefficients into Equation 2 then allows us to compute loss ($L$) as a function of parameters and data:

$$L(N, D) = 1.87 + \frac{521}{N^{0.353}} + \frac{1488}{D^{0.353}} \tag{20}$$

To verify the accuracy of our fit, we benchmark the predictions with those of the IsoFLOP C4 curves in [42]. Following [42], we can compute the optimal number of parameters $N_{opt}$ and tokens $D_{opt}$ for our fit using:

$$N_{opt}(C) = G\left(\frac{C}{6}\right)^a, \quad D_{opt}(C) = G^{-1}\left(\frac{C}{6}\right)^b \tag{21}$$

$$\text{where} \quad G = \left(\frac{\alpha A}{\beta B}\right)^{\frac{1}{\alpha+\beta}}, \quad a = \frac{\beta}{\alpha + \beta}, \text{ and } b = \frac{\alpha}{\alpha + \beta}$$

Given the Gopher compute budget of $C = 5.76 \times 10^{23}$ our fitted parameters predict an optimal allocation of $N_{opt} = 70.0$ billion parameters and $D_{opt} = 1.37$ trillion tokens. This is very close to the 73 billion parameters and 1.3 trillion tokens predicted by the IsoFLOP curves on C4 from [42] and thus we consider it a good fit. We use these fitted parameters rather than the MassiveWeb parameters for all computations involving Chinchilla scaling laws.

## C  Additional Contour Plots

Figure 8 contains additional empirical isoLoss contours for 400 million and 1.5 billion unique tokens. Results show that like in Figure 3 significantly lower loss can be achieved by increasing parameters and epochs beyond what is compute-optimal at a single epoch. The lowest loss is also achieved by allocating more extra compute to repeating data rather than to adding parameters.

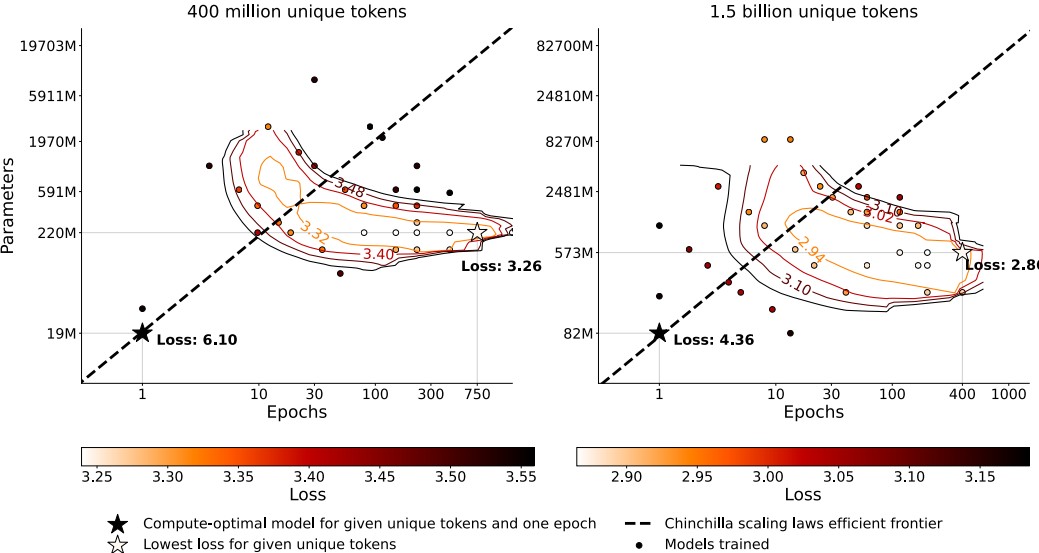

Figure 8: **Empirical isoLoss curves for 400 million and 1.5 billion unique tokens.** 34 models trained on 400 million unique tokens and 37 models trained on 1.5 billion unique tokens with varying parameters and epochs.

## D  Double Descent

Prior work has reported double descent phenomena when repeating data, where the loss initially increases and then decreases again as the model is trained for more epochs [75, 40]. In Figure 9, we plot the loss curves of several models trained for varying epochs on 100 million tokens. We find double descent phenomena with the loss of all models increasing at 200 epochs before decreasing

again. This contributes to additional noise in the fitting of our functions in Appendix A, as our functional form assumes loss to be monotonically decreasing as epochs increase. Thus, we remove most such examples from the fitting.

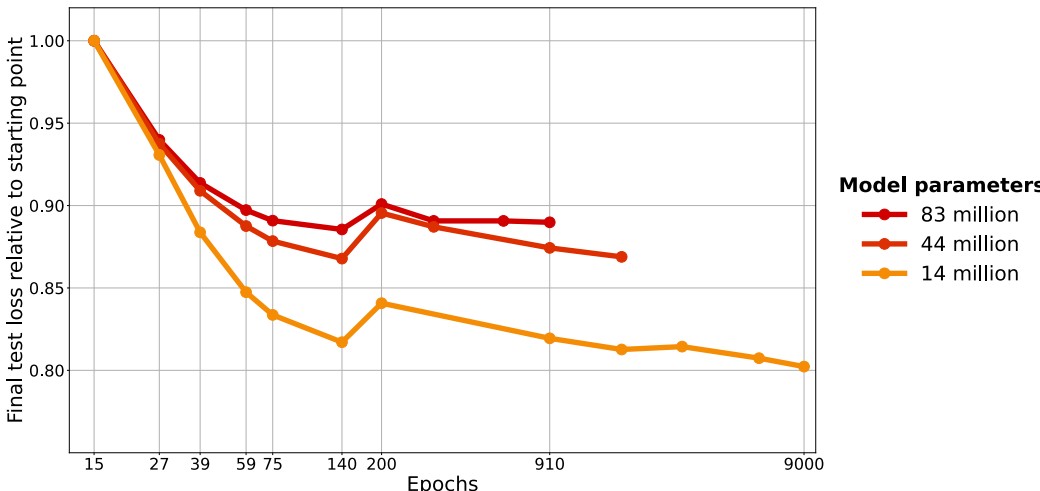

Figure 9: **Double descent.** Each dot is a model trained on 100 million unique tokens. Loss initially increases at 200 epochs and then decreases again; this is known as epoch-wise double descent [75].

# E    Repeating on Heavily Deduplicated Data

To investigate whether Figure 3 is dependent on the inherent amount of duplicates in the selected 100 million tokens, we train several models on a deduplicated version of C4 (see Appendix N). We plot the performance of the models trained on the deduplicated C4 versus the regular C4 in Figure 10. All models are evaluated on the same validation dataset from the regular C4. Regardless of deduplication we find 59 epochs to be optimal and the overall trend to be very similar. Together with our results on OSCAR (Appendix I), this suggests that our work generalizes to different datasets with different inherent amounts of duplicates.

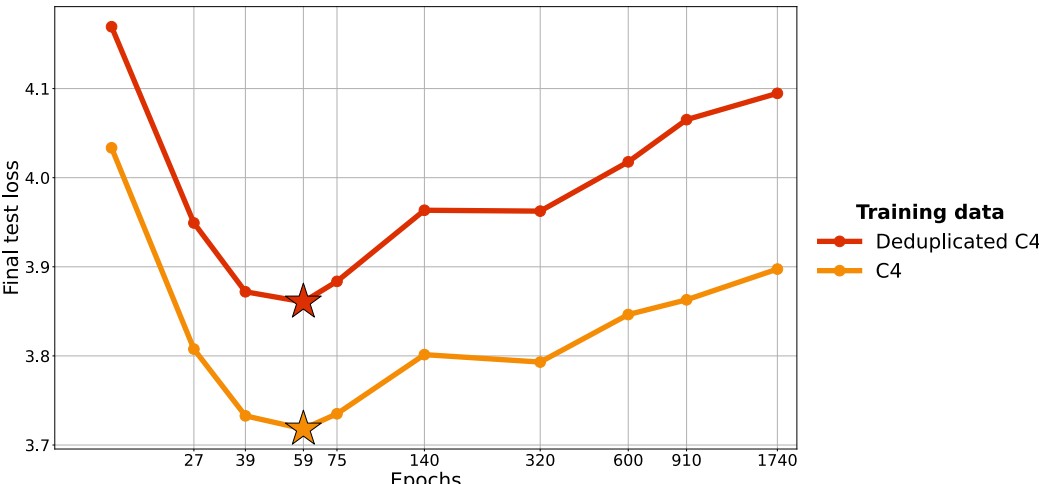

Figure 10: **Optimal loss on deduplicated data.** 146 million parameter models trained on 100 million unique tokens that are either directly from C4 or undergo additional deduplication. Each dot is a single model. While deduplication results in a higher test loss, the optimal number of epochs remains the same whether or not deduplication is performed (see also Figure 3).

# F  Do Excess Parameters Hurt, Plateau or Help?

Figures 3, 8 suggest that excess parameters (or epochs) can harm performance. We hypothesize that this is due to suboptimal hyperparameters and could be prevented with better regularization. Thus, we expect with optimal regularization hyperparameters excess parameters would never hurt, but performance would merely plateau, as in extreme cases regularization could just take the form of removing the excess parameters. One approach to selecting optimal hyperparameters is $\mu$P [127]. We compare excessively large models trained with a data constraint of $D_C = 100$ million tokens in Figure 11 across $\mu$P, our default hyperparameters (Appendix S) and scaling law predictions. Surprisingly, $\mu$P leads to even higher test loss than our default hyperparameters. Nevertheless, we find that also with $\mu$P excessive parameters hurt: The models with more than 2 billion parameters have significantly higher validation loss after training than the models with 200 million to 1 billion parameters when trained on only 100 million tokens. However, $\mu$P only covers hyperparameters such as the learning rate, but not explicit regularization hyperparameters like dropout rates, which we hypothesize would prevent this behavior. Thus, our proposed scaling equations predict loss to plateau, as seen in the straight line. As the compute-optimal parameter count for 100 million tokens is around 7 million, all depicted models have a significant amount of excess parameters and data-constrained scaling laws predict their losses to be all the same ($R_N^* \ll R_N$). Meanwhile, the default Chinchilla scaling law [42] predicts loss to continue decreasing as parameters are added, which is in stark contrast to the empirical data.

If one wants to incorporate excess parameters hurting performance into the scaling law equations, one could consider **(a)** Modifying the exponential decay formulation introduced in Appendix A such that instead of the value of repeated data decaying to 0 it decays to a large negative value **(b)** decaying the exponents $\alpha$ and $\beta$ in Equation 7 instead of $D$ and $N$. Decaying the exponents to 0 has the effect of more repetitions eventually hurting performance as $\lim_{\alpha \to 0} D^\alpha = 1$ and the same for $\beta$. Thus, initially as $D$ and $N$ increase loss decreases, but ultimately the decay of $\alpha$ and $\beta$ pushes $D$ and $N$ back to 1 resulting in loss to increase. Specifically, approach **(b)** could take the form of:

$$L(N, D, R_N, R_D) = E + \frac{A}{N^{\alpha * max(0, 1-(R_N/R_N^*))}} + \frac{B}{D^{\beta * max(0, 1-(R_D/R_D^*))}} \tag{22}$$

Like the equations in Appendix A this formulation also reduces to the Chinchilla scaling laws in the base case of $R_D = 0$ or $R_N = 0$. As the exponents decrease with more repetitions adding parameters or epochs becomes less beneficial. Eventually, the decay in $\alpha$ or $\beta$ causes loss to increase again as it pushes $N$ or $D$ back down to 1. We fit this formula using the same approach outlined in Appendix A but including samples where excess parameters or epochs hurt (296 total samples). We use a grid of initialization given by: $R_N^* \in \{0., 2000., \ldots, 100000.\}$ and $R_D^* \in \{0., 2000., \ldots, 100000.\}$. This results in $R_D^* = 26530.611$ and $R_N^* = 2040.8163$. $R_N^*$ is significantly lower resulting in excess parameters hurting faster than excess epochs, which is in line with empirical data from Figure 3. We visualize Figure 3 with the predictions from this alpha-beta decay formulation in Figure 12. Expected parameters eventually hurt resulting in circle-shaped contours. Due to the very high $R_D^*$ the area where epochs start to hurt is outside of the boundaries of Figure 12. While the predicted optimal allocation (efficient frontier) is similar to Figure 3, the predicted return from repeated data differs significantly. The alpha-beta decay formulation incorrectly predicts returns to diminish significantly slower as seen by the longer efficient frontier and the smaller distance in contours early on as compared to Figure 3. Beyond its potentially useful properties, we do not have a rigorous mathematical justification for this alpha-beta decay formulation which could be the cause of the incorrect return predictions.

Ultimately, we settle on our exponential decay formulation from Appendix A that does not allow excess parameters or epochs to hurt, as preventing such behavior is trivial by stopping training (in the case of epochs hurting) or removing excess parameters (in the case of model parameters hurting). Further, accurately predicting how much loss increases in the limit is not very useful, as in practice one would want to stop training when it's expected to plateau anyways.

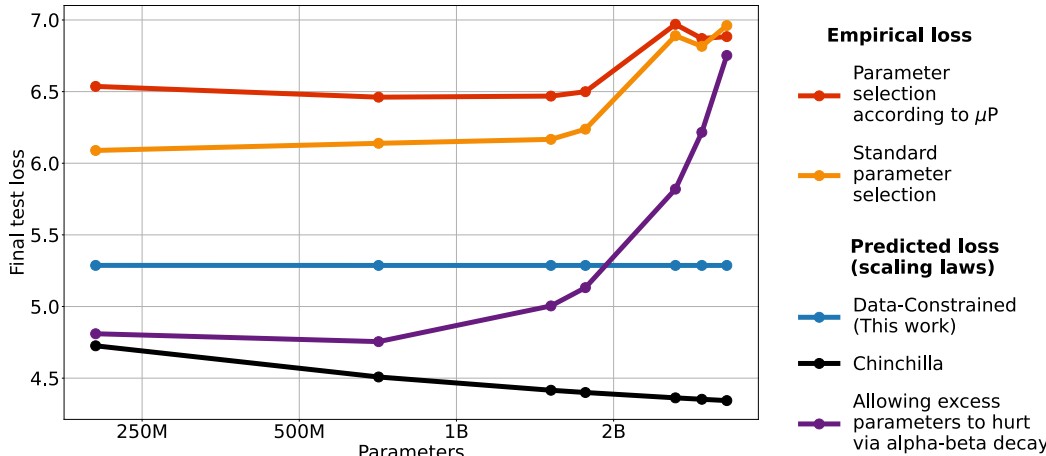

Figure 11: **Empirical and predicted losses of LLMs trained on 100 million tokens for a single epoch.** Excess parameters empirically hurt performance, but this may be due to a lack of regularization. Thus, our scaling formula predicts loss to plateau, while Chinchilla predicts loss to improve. By decaying the exponent $\alpha$ (and $\beta$) instead, one can allow excess parameters to hurt.

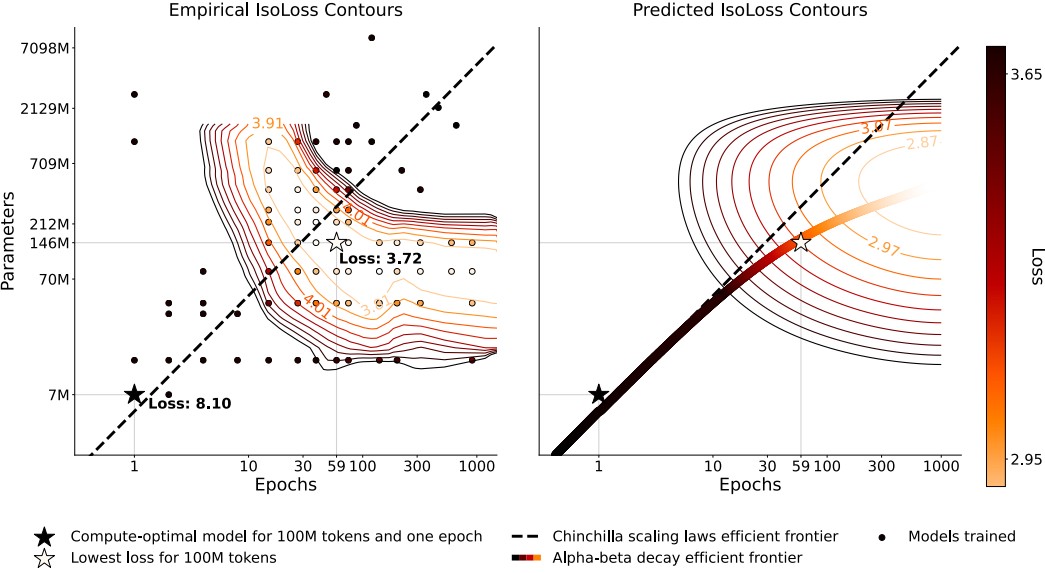

Figure 12: **IsoLoss contours for 100 million unique tokens with contours predicted by parametric decay of alpha and beta.** The same models from Figure 3 with the contour predictions being done by the alpha-beta decay formulation introduced in Appendix F.

# G   Case Study: Galactica

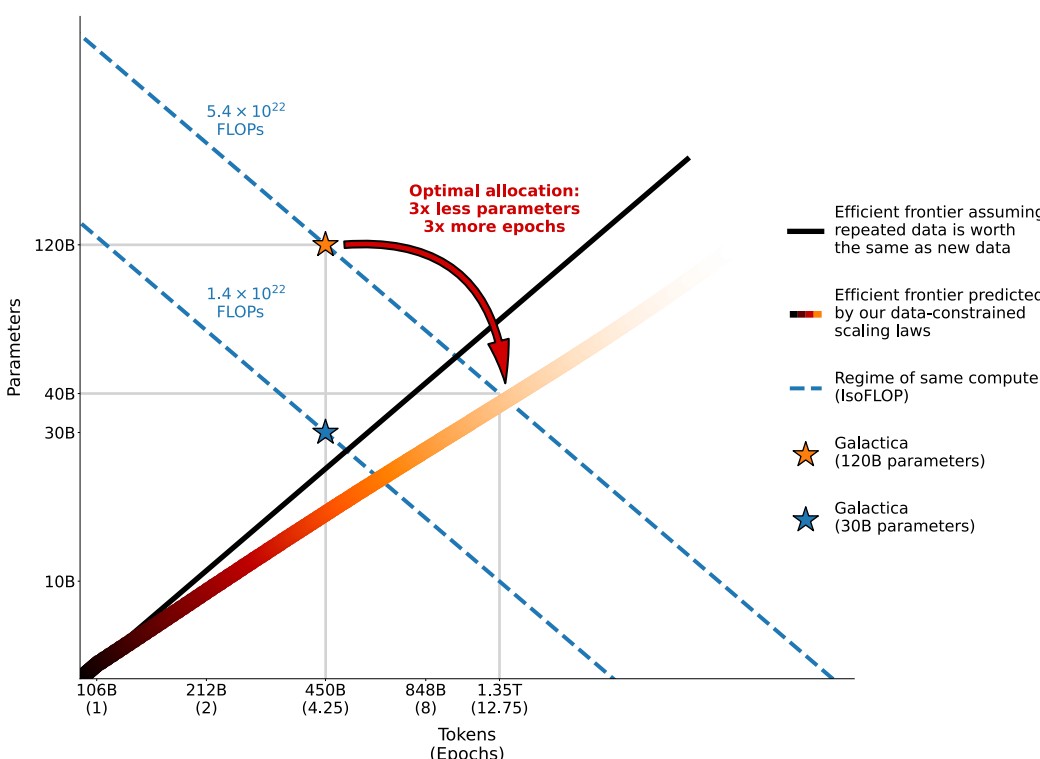

Figure 13: **Optimal compute allocation for Galactica.** Efficient frontier assuming repeated data is worth the same as new data (Chinchilla scaling laws) and data-constrained efficient frontier assuming a unique token budget of 106 billion tokens like for the Galactica models [108]. For optimal compute allocation according to our proposed data-constrained scaling laws, the 120 billion Galactica model should have been significantly smaller and trained for more epochs.

The Galactica models [108] are the only publicly known LLMs that explicitly trained for a significant number of epochs prior to this work. They trained their models on 106 billion unique tokens for 4.25 epochs. Our findings on *Return* from repeated data agree with their conclusion that multiple epochs are beneficial, however, we find that even more epochs can be beneficial and a small spike in validation loss does not justify stopping training (Appendix J). Meanwhile, our findings on *Allocation* significantly deviate from Galactica. Figure 13 visualizes the Galactica models with our predicted efficient frontier in the same style as Figure 1. The creators of Galactica decided to train a 120 billion parameter model on 450 billion tokens, a significant overallocation to parameters even in Chinchilla terms (black efficient frontier). This decision was likely driven by the intuition that repeated data is worth less, thus one should spend more compute on parameters. However, our empirical data contradicts this. Parameters learning from repeated data are worth *even less* than repeated data, thus one should overallocate to epochs, not parameters. Our data-constrained scaling laws thus predict that a better model could have been trained by allocating significantly more FLOPs to epochs rather than parameters for the largest Galactica model with 120 billion parameters. Specifically, 40 billion parameters trained for 1.35 trillion tokens (12.75 epochs) would have been optimal according to data-constrained scaling laws. Note that these scaling laws have been fitted on C4, which is not the dataset used to pre-train Galactica. The Galactica models are pre-trained on a predominantly scientific dataset, which includes code data among other data sources. Results from [42] show that there are differences in the scaling coefficients when training on C4 as compared to GitHub code, however, the overall allocation trend is the same. Thus, while we expect a smaller model trained for more epochs to be better than the 120 billion parameter model, the optimal allocation is unlikely to be exactly 40 billion parameters and 1.35 trillion tokens.

## H Training Loss

Hoffmann et al. [42] use training loss as their core metric. However, when repeating data for multiple epochs, training loss is a bad metric as models will overfit to the limited data available as shown in Figure 14. Thus, we use loss on a held-out test set as our key performance metric.

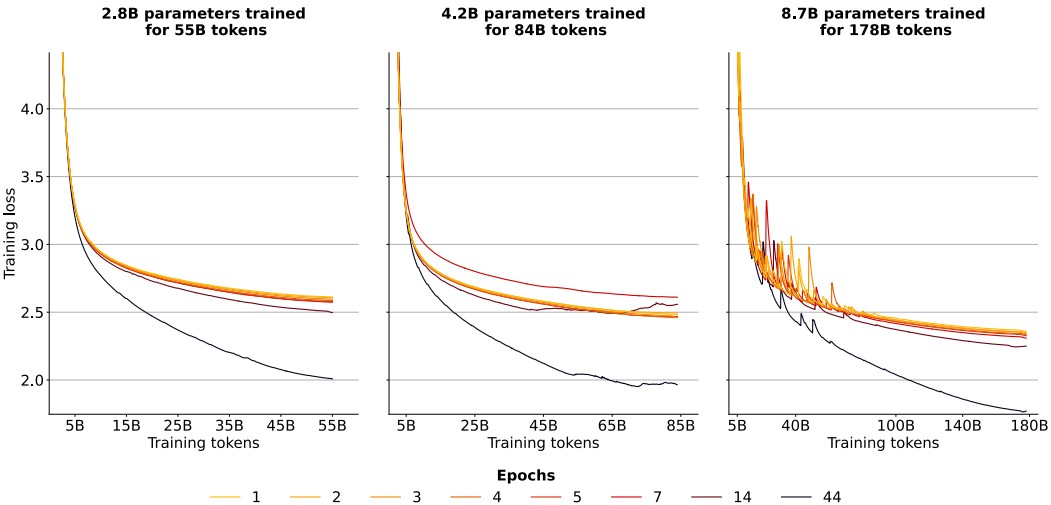

Figure 14: **Training loss smoothed with exponential moving average smoothing and a weight of 0.999.** Models trained on fewer unique tokens (more epochs) have better training loss as they overfit.

## I Scaling Curves on the OSCAR Corpus

To ensure our findings are not dataset-dependent, we train models with the same configurations from Figure 4 on the OSCAR corpus [83]. OSCAR is considered noisier than C4 [90] due to its less stringent duplication. Figures 15,16 depict the validation and training loss of these models. We find the trend to be the same as for models trained on C4: While models with fewer repeats have better loss, differences for a few repeats are insignificant.

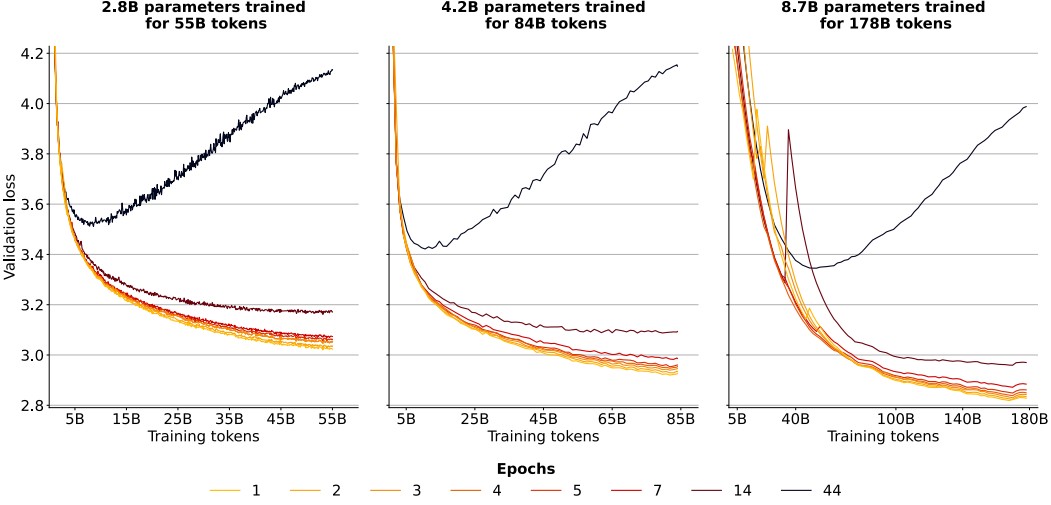

Figure 15: **Validation loss during training for models trained on OSCAR.** Models trained on tokens that are repeated for multiple epochs have consistently worse loss.

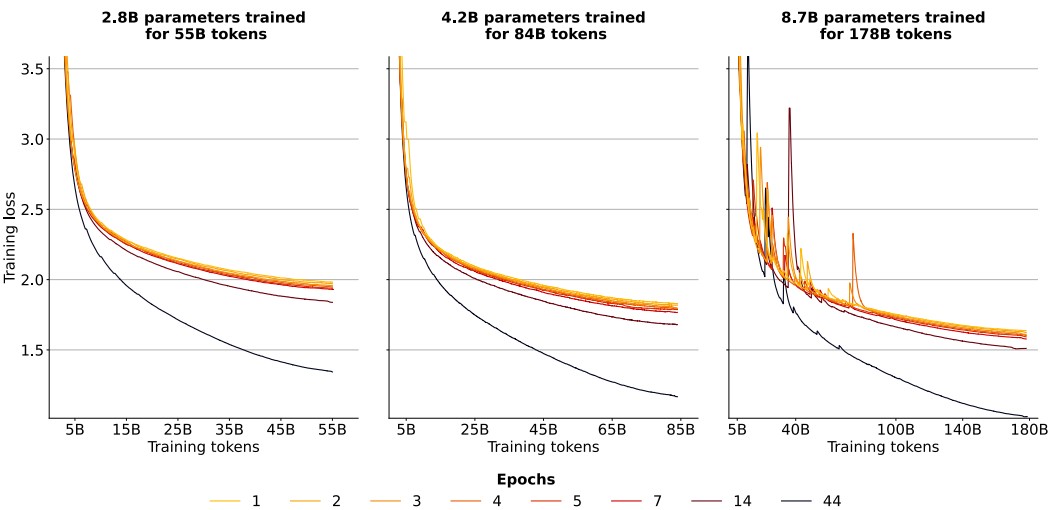

Figure 16: **Training loss for models trained on OSCAR smoothed with exponential moving average smoothing and a weight of 0.999.** Models trained on fewer unique tokens (more epochs) have better training loss as they overfit.

## J   Validation Loss by Epoch

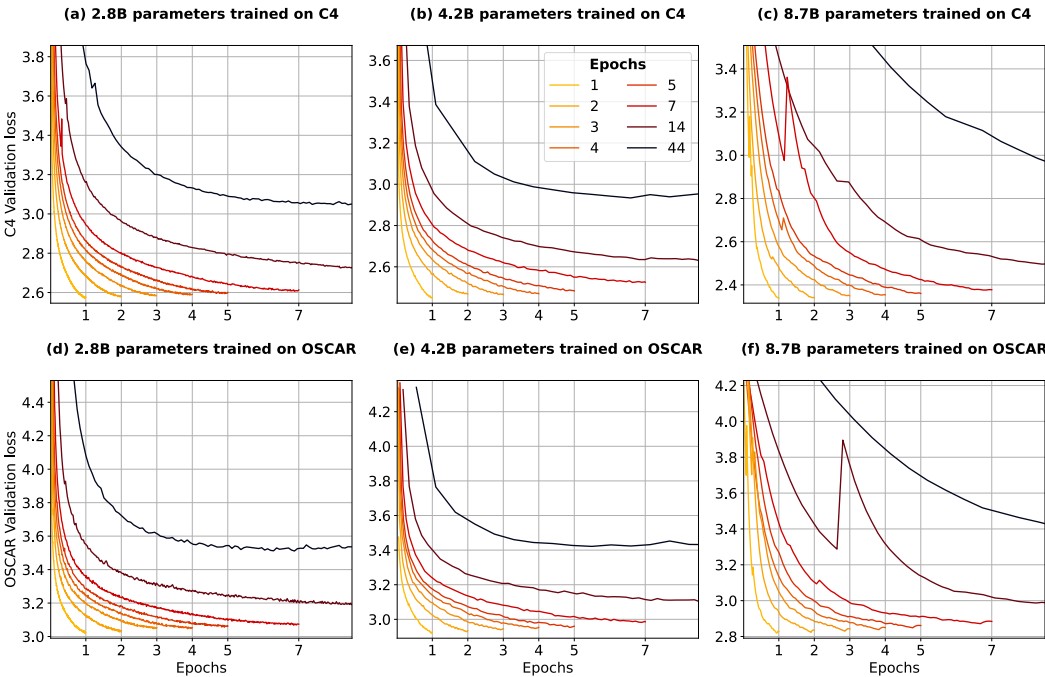

Figure 17: **Validation loss during training visualized by epochs.** Loss progresses smoothly throughout training. There are temporary spikes for 8.7 billion parameter models, commonly at the start of a new epoch.

Taylor et al. [108] decided to early-stop pre-training of the Galactica models due to a small increase in validation loss at the start of the fifth epoch. In Figure 17 we plot the validation loss curves of our isoFLOP models as a function of epochs. We do find small increases in validation loss when models enter a new epoch. For example, upon entering the third and fourth epoch, the 7-epoch 8.7 billion

parameter OSCAR model shows loss spikes. However, these are temporary and loss continues to go down smoothly thereafter. Thus, we hypothesize that the Galactica models could have attained better performance by continuing pre-training beyond the loss spike experienced at the beginning of the fifth epoch.

# K    Evaluation Details

Table 2: **Setup for computing validation loss during training.** At every *Evaluation Interval*, loss is computed on *Evaluation Tokens* many tokens from the validation set. The evaluation tokens vary with the interval, i.e. the evaluation tokens at 100 steps are not the same as at 200 steps. However, the tokens do not vary across data budgets for the same FLOP budget (Figure 4). For example, $N = 2.8$ billion parameter models with $D_C = 55$ billion tokens are evaluated on the same data as models with $D_C = 28$ billion tokens at each evaluation interval.

| FLOP budget | Parameters | Evaluation Interval | Evaluation Tokens |
|---|---|---|---|
| $9.3 \times 10^{20}$ | 2.8B | 100 | 105 million |
| $2.1 \times 10^{21}$ | 4.2B | 1000 | 105 million |
| $9.3 \times 10^{21}$ | 8.7B | 1000 | 2.1 million |

**Loss evaluation**    For all models trained on C4, the final test loss is computed on the same 210 million tokens from the C4 validation set after training. For held-out evaluation during training, such as in Figure 4, the configurations are displayed in Table 2. The small number of evaluation tokens for the 8.7 billion parameter models likely contributes to the loss spikes for 8.7 billion parameter models seen in Figure 4. Thus, we smooth the validation loss curves of 8.7 billion parameter models with exponential moving average smoothing and a weight of 0.85. For training OSCAR, configurations are the same, however, the validation split used is a held-out part from the OSCAR training split, as there is no official validation split for OSCAR. All training loss curves for C4 and OSCAR models are smoothed with exponential moving average smoothing and a weight of 0.999.

**Downstream evaluation**    We provide statistics of all downstream evaluation datasets in Table 3. We use the evaluation-harness frameworks from BigScience and EleutherAI [32] to evaluate models on 19 evaluation datasets. For each dataset, a maximum of 3000 samples are evaluated with 0,1,2,3,4 and 5 few-shots [15] to produce six scores which are then averaged. We normalize scores to range from the random baseline of each task to 1 and report them as percentages. For example, if random guessing produces 50% accuracy and the maximum accuracy possible is 100%, then a raw accuracy of 55% would be normalized to 10%, and a raw accuracy of 45% would be normalized to -10% since it is worse than random. This is done to give all tasks the same weight. Otherwise average performance would heavily depend on generative tasks, where the random baselines are 0. Prompts are sourced from GPT-3 [15] and PromptSource [5] and detailed in Appendix T. We note that our evaluation is in no means comprehensive and a larger benchmarking would be helpful [102, 73]. However, by training five seeds for most models benchmarked, always averaging 0-5 fewshots, and ensuring maximum data overlap for repeated data (§4) we significantly reduce uncertainty.

Table 3: **Downstream evaluation datasets.** We evaluate on 19 datasets: The first 14 are evaluated using accuracy (ANLI counted as three), the next 4 using ROUGE-2 f-measure [59] and bAbI using exact match.

| Dataset | Split(s) | Samples | Baseline | URL |
|---|---|---|---|---|
| ANLI [79] | dev_r1,2,3 | 3000 | 33.3 | `hf.co/datasets/anli` |
| ARC-Easy [22] | test | 1172 | 25.0 | `hf.co/datasets/ai2_arc` |
| ARC-Challenge [22] | test | 2376 | 25.0 | `hf.co/datasets/ai2_arc` |
| BoolQ [21] | validation | 3270 | 50.0 | `hf.co/datasets/boolq` |
| CB [25] | validation | 56 | 33.3 | `hf.co/datasets/super_glue` |
| Copa [92] | validation | 100 | 50.0 | `hf.co/datasets/super_glue` |
| HellaSwag [129] | test | 10003 | 25.0 | `hf.co/datasets/hellaswag` |
| PiQA [12] | validation | 1838 | 50.0 | `hf.co/datasets/piqa` |
| RTE [24, 114] | validation | 277 | 50.0 | `hf.co/datasets/super_glue` |
| SciQ [120] | test | 1000 | 25.0 | `hf.co/datasets/sciq` |
| StoryCloze 2016 [69] | test | 1871 | 25.0 | `hf.co/datasets/story_cloze` |
| WinoGrande XL [93] | test | 1267 | 50.0 | `hf.co/datasets/winogrande` |
| E2E NLG [29] | test | 4693 | 0.0 | `hf.co/datasets/e2e_nlg_cleaned` |
| XSUM [77, 34] | test | 11334 | 0.0 | `hf.co/datasets/GEM/xsum` |
| WebNLG EN [17, 34] | test | 5150 | 0.0 | `hf.co/datasets/GEM/web_nlg` |
| WikiLingua EN [53, 34] | sampled_test | 3000 | 0.0 | `hf.co/datasets/GEM/wiki_lingua` |
| bAbI [122] | test | 19000 | 0.0 | `hf.co/datasets/Muennighoff/babi` |

## L  Downstream Repetition Results

In Tables 4-9 we report downstream results of all models trained on C4 [90] and OSCAR [83] according to the configurations in Figure 4. All scores are from the final checkpoints at the end of training. OSCAR is a noisier dataset than C4 due to less filtering, thus models trained on C4 generally perform better. Notably, models trained on C4 completely fail on bAbI [122], while OSCAR models are able to perform better than random. This is likely due to code data being present in OSCAR, which enables state-tracking capabilities like for code augmented models in §7. For C4 the creators strictly removed all data that resembles code [90]. There are no significant differences between models trained for a single epoch and models trained for up to 4 epochs. Even models trained for more epochs (and thus on less unique data) have similar performance.

Table 4: **Results for 2.8B parameter models trained on repeated data on C4 for 55B total tokens.** Scores are normalized averages of 0-5 few-shots and reported as percentages. We report mean/std. err. across five different models, each trained with a different random seed.

| Data Budget | 55B | 28B | 18B | 14B | 11B | 9B | 4B | 1.25B |
|---|---|---|---|---|---|---|---|---|
| Epochs | 1 | 2 | 3 | 4 | 5 | 7 | 14 | 44 |
| ANLI R1 | 0.4 ± 1.6 | 0.7 ± 0.8 | 0.3 ± 0.5 | -0.3 ± 1.8 | 0.4 ± 1.8 | 0.4 ± 0.7 | 0.0 ± 0.9 | -0.6 ± 0.6 |
| ANLI R2 | 0.9 ± 0.4 | 1.4 ± 0.8 | 0.8 ± 0.8 | 1.1 ± 0.7 | 0.5 ± 0.7 | 0.6 ± 1.0 | 1.1 ± 1.1 | 2.7 ± 1.6 |
| ANLI R3 | 1.7 ± 0.5 | 1.2 ± 0.4 | 0.4 ± 0.5 | 1.9 ± 0.7 | 0.6 ± 1.0 | 0.8 ± 0.8 | 1.7 ± 0.7 | 0.7 ± 1.7 |
| ARC-Challenge | 1.6 ± 1.0 | 0.9 ± 0.5 | 1.2 ± 0.6 | 1.1 ± 0.6 | 1.1 ± 1.2 | 1.3 ± 0.5 | 0.3 ± 0.6 | -2.9 ± 1.0 |
| ARC-Easy | 44.5 ± 0.5 | 44.9 ± 0.4 | 44.7 ± 0.7 | 44.3 ± 0.4 | 44.0 ± 0.5 | 44.2 ± 0.9 | 41.4 ± 0.2 | 28.9 ± 0.7 |
| BoolQ | 18.8 ± 3.4 | 16.2 ± 5.2 | 16.1 ± 2.7 | 19.7 ± 1.8 | 15.0 ± 3.8 | 16.9 ± 3.2 | 13.1 ± 4.9 | -2.1 ± 4.7 |
| CB | 20.0 ± 4.7 | 17.4 ± 6.4 | 14.6 ± 5.1 | 17.5 ± 4.2 | 12.3 ± 12.2 | 14.4 ± 7.5 | 21.6 ± 8.4 | 21.3 ± 5.6 |
| COPA | 49.7 ± 3.5 | 50.3 ± 3.4 | 49.9 ± 2.3 | 50.1 ± 2.5 | 50.9 ± 1.2 | 48.1 ± 2.4 | 43.5 ± 3.1 | 33.3 ± 1.9 |
| HellaSwag | 24.7 ± 0.3 | 24.6 ± 0.2 | 24.3 ± 0.1 | 24.3 ± 0.0 | 24.3 ± 0.3 | 24.1 ± 0.1 | 22.8 ± 0.2 | 16.7 ± 0.4 |
| PiQA | 47.9 ± 0.6 | 47.6 ± 0.8 | 47.3 ± 0.3 | 47.6 ± 0.6 | 47.6 ± 0.7 | 47.0 ± 0.2 | 45.6 ± 0.5 | 37.0 ± 0.4 |
| RTE | 5.1 ± 4.0 | 2.5 ± 4.5 | 8.4 ± 2.6 | 6.0 ± 2.5 | 5.1 ± 1.6 | 2.3 ± 3.9 | 7.8 ± 2.5 | 2.6 ± 4.3 |
| SciQ | 83.2 ± 0.6 | 82.5 ± 0.6 | 82.7 ± 1.1 | 81.9 ± 0.6 | 81.9 ± 0.8 | 81.6 ± 0.9 | 78.5 ± 1.1 | 59.3 ± 1.6 |
| StoryCloze 2016 | 58.7 ± 0.2 | 58.7 ± 0.5 | 58.5 ± 0.3 | 58.3 ± 0.3 | 58.5 ± 0.6 | 58.4 ± 0.3 | 56.7 ± 0.5 | 52.0 ± 0.6 |
| WinoGrande XL | 11.6 ± 0.8 | 10.8 ± 1.1 | 10.9 ± 1.3 | 10.6 ± 0.5 | 11.1 ± 0.9 | 10.6 ± 0.9 | 6.4 ± 1.3 | 2.9 ± 1.3 |
| E2E NLG | 17.0 ± 1.4 | 17.7 ± 0.5 | 17.0 ± 1.2 | 16.9 ± 1.1 | 15.1 ± 2.3 | 13.3 ± 2.2 | 14.9 ± 0.9 | 9.8 ± 0.9 |
| XSUM | 2.4 ± 0.1 | 2.4 ± 0.1 | 2.5 ± 0.1 | 2.3 ± 0.2 | 2.4 ± 0.1 | 2.4 ± 0.1 | 2.1 ± 0.1 | 1.6 ± 0.1 |
| WebNLG EN | 5.3 ± 0.1 | 5.5 ± 0.2 | 5.4 ± 0.1 | 5.4 ± 0.1 | 5.1 ± 0.1 | 5.4 ± 0.2 | 5.1 ± 0.3 | 2.9 ± 0.2 |
| WikiLingua EN | 3.0 ± 0.1 | 3.1 ± 0.1 | 2.9 ± 0.1 | 2.9 ± 0.3 | 2.9 ± 0.2 | 2.9 ± 0.1 | 2.6 ± 0.1 | 2.0 ± 0.2 |
| bAbI | 0.0 ± 0.0 | 0.0 ± 0.0 | 0.0 ± 0.0 | 0.0 ± 0.0 | 0.0 ± 0.0 | 0.0 ± 0.0 | 0.0 ± 0.0 | 0.0 ± 0.0 |
| Average | 20.9 ± 0.4 | 20.4 ± 0.3 | 20.4 ± 0.2 | 20.6 ± 0.2 | 19.9 ± 0.9 | 19.7 ± 0.2 | 19.2 ± 0.5 | 14.1 ± 0.4 |

Table 5: **Results for 2.8B parameter models trained on repeated data on OSCAR for 55B total tokens.** Scores are normalized averages of 0-5 few-shots and reported as percentages. We report mean/std. err. across five different models, each trained with a different random seed.

| Data Budget | 55B | 28B | 18B | 14B | 11B | 9B | 4B | 1.25B |
|---|---|---|---|---|---|---|---|---|
| Epochs | 1 | 2 | 3 | 4 | 5 | 7 | 14 | 44 |
| ANLI R1 | -0.3 ± 0.5 | -0.6 ± 1.3 | 0.2 ± 0.6 | 0.3 ± 1.1 | 0.2 ± 1.2 | -0.1 ± 1.1 | -0.1 ± 0.5 | -0.7 ± 1.3 |
| ANLI R2 | 1.0 ± 1.0 | 1.1 ± 0.3 | 1.7 ± 0.8 | 2.3 ± 0.8 | 1.4 ± 1.0 | 0.8 ± 0.7 | 1.0 ± 0.7 | 2.3 ± 0.7 |
| ANLI R3 | 0.4 ± 0.8 | 0.5 ± 0.5 | -0.2 ± 0.8 | -0.1 ± 1.0 | 1.1 ± 0.6 | 0.7 ± 0.4 | -0.2 ± 0.9 | 0.5 ± 1.2 |
| ARC-Challenge | -1.4 ± 0.8 | -0.6 ± 0.8 | -1.7 ± 0.1 | -1.6 ± 0.7 | -1.6 ± 0.6 | -1.4 ± 0.5 | -1.9 ± 0.8 | -5.0 ± 1.1 |
| ARC-Easy | 39.7 ± 0.3 | 39.6 ± 0.8 | 39.5 ± 0.6 | 39.3 ± 0.5 | 38.7 ± 0.6 | 38.7 ± 0.4 | 36.9 ± 0.4 | 25.4 ± 0.7 |
| BoolQ | 12.8 ± 4.4 | 7.8 ± 3.8 | 7.9 ± 3.8 | 3.3 ± 5.4 | 0.2 ± 3.0 | 2.3 ± 6.1 | -2.1 ± 2.4 | 7.4 ± 6.1 |
| CB | 19.7 ± 5.1 | 15.4 ± 7.3 | 13.2 ± 5.1 | 12.6 ± 2.6 | 21.7 ± 3.6 | 15.4 ± 3.7 | 16.2 ± 5.2 | 9.7 ± 5.7 |
| COPA | 42.7 ± 2.2 | 39.5 ± 2.2 | 40.9 ± 2.0 | 41.5 ± 2.1 | 38.5 ± 2.4 | 40.4 ± 2.4 | 38.6 ± 2.6 | 28.5 ± 3.1 |
| HellaSwag | 16.3 ± 0.1 | 16.3 ± 0.2 | 16.3 ± 0.2 | 16.1 ± 0.2 | 16.0 ± 0.1 | 15.9 ± 0.2 | 15.0 ± 0.2 | 11.7 ± 0.1 |
| PiQA | 41.2 ± 0.7 | 41.4 ± 0.5 | 40.3 ± 0.4 | 40.6 ± 0.5 | 40.3 ± 0.9 | 39.8 ± 0.6 | 38.8 ± 1.1 | 31.0 ± 0.4 |
| RTE | 3.9 ± 1.1 | 2.1 ± 1.6 | 2.3 ± 3.3 | 1.6 ± 3.0 | 0.5 ± 2.1 | 2.9 ± 2.5 | 0.9 ± 3.4 | -3.2 ± 2.7 |
| SciQ | 83.2 ± 0.6 | 82.4 ± 0.6 | 82.1 ± 0.9 | 82.6 ± 0.7 | 81.5 ± 0.9 | 80.5 ± 0.6 | 76.5 ± 1.3 | 57.7 ± 1.8 |
| StoryCloze 2016 | 52.8 ± 0.3 | 52.9 ± 0.4 | 52.6 ± 0.3 | 53.0 ± 0.4 | 52.3 ± 0.4 | 52.4 ± 0.4 | 51.8 ± 0.7 | 47.9 ± 0.5 |
| WinoGrande XL | 5.8 ± 0.9 | 4.4 ± 1.4 | 4.5 ± 0.3 | 4.2 ± 1.3 | 4.5 ± 0.6 | 4.1 ± 0.7 | 1.7 ± 1.2 | 0.8 ± 1.3 |
| E2E NLG | 20.3 ± 0.3 | 19.9 ± 0.5 | 19.9 ± 0.7 | 20.9 ± 0.9 | 19.7 ± 0.7 | 20.4 ± 0.6 | 19.1 ± 0.8 | 14.2 ± 0.7 |
| XSUM | 3.0 ± 0.1 | 2.9 ± 0.0 | 2.9 ± 0.3 | 2.9 ± 0.2 | 2.9 ± 0.1 | 2.8 ± 0.3 | 2.6 ± 0.2 | 1.8 ± 0.1 |
| WebNLG EN | 8.8 ± 0.4 | 8.3 ± 0.6 | 8.5 ± 0.3 | 8.4 ± 0.6 | 8.1 ± 0.2 | 8.2 ± 0.2 | 7.2 ± 0.3 | 3.3 ± 0.3 |
| WikiLingua EN | 2.9 ± 0.1 | 3.1 ± 0.2 | 3.1 ± 0.1 | 3.0 ± 0.1 | 3.1 ± 0.1 | 3.2 ± 0.3 | 2.7 ± 0.2 | 1.7 ± 0.2 |
| bAbI | 15.5 ± 1.0 | 15.7 ± 1.1 | 15.3 ± 0.8 | 15.1 ± 1.5 | 15.9 ± 1.1 | 16.2 ± 0.9 | 14.3 ± 0.6 | 6.6 ± 0.6 |
| Average | 19.4 ± 0.5 | 18.5 ± 0.2 | 18.4 ± 0.4 | 18.2 ± 0.4 | 18.2 ± 0.4 | 18.1 ± 0.4 | 16.8 ± 0.5 | 12.7 ± 0.7 |

Table 6: **Results for 4.2B parameter models trained on repeated data on C4 for 84B total tokens.** Scores are normalized averages of 0-5 few-shots and reported as percentages. We report mean/std. err. across five different models, each trained with a different random seed.

| Unique Tokens | 84B | 42B | 28B | 21B | 17B | 12B | 6B | 1.9B |
|---|---|---|---|---|---|---|---|---|
| Epochs | 1 | 2 | 3 | 4 | 5 | 7 | 14 | 44 |
| ANLI R1 | -1.0 ± 0.3 | -0.7 ± 1.1 | -0.7 ± 1.0 | -0.4 ± 1.1 | 0.4 ± 0.8 | 0.5 ± 1.1 | 0.1 ± 0.9 | 0.2 ± 0.9 |
| ANLI R2 | 0.8 ± 0.5 | 0.8 ± 0.8 | 0.0 ± 1.4 | 0.5 ± 0.7 | 0.5 ± 0.9 | 0.3 ± 1.0 | 0.7 ± 0.7 | 2.5 ± 1.0 |
| ANLI R3 | 1.1 ± 0.7 | 0.8 ± 0.9 | 0.3 ± 0.8 | 1.4 ± 1.1 | 1.3 ± 0.9 | 2.3 ± 0.2 | 1.3 ± 0.2 | 1.6 ± 1.2 |
| ARC-Challenge | 5.3 ± 0.6 | 5.1 ± 0.9 | 5.2 ± 2.0 | 6.0 ± 0.8 | 4.7 ± 0.8 | 3.1 ± 0.4 | 2.9 ± 1.0 | -1.3 ± 1.0 |
| ARC-Easy | 49.2 ± 0.9 | 50.4 ± 1.2 | 47.4 ± 4.5 | 49.4 ± 0.7 | 48.7 ± 1.5 | 44.9 ± 0.7 | 45.0 ± 1.2 | 31.9 ± 0.9 |
| BoolQ | 18.2 ± 4.0 | 19.6 ± 5.1 | 22.1 ± 1.0 | 20.4 ± 3.6 | 18.4 ± 6.0 | 18.4 ± 3.9 | 18.9 ± 2.6 | -3.3 ± 7.1 |
| CB | 12.0 ± 7.2 | 8.5 ± 9.2 | 7.9 ± 10.4 | 19.6 ± 7.3 | 17.8 ± 7.3 | 15.1 ± 5.8 | 17.5 ± 3.5 | 19.5 ± 6.6 |
| COPA | 59.1 ± 5.4 | 57.7 ± 3.5 | 56.7 ± 2.0 | 55.5 ± 2.4 | 56.8 ± 1.8 | 58.9 ± 1.7 | 48.7 ± 3.3 | 34.9 ± 3.4 |
| HellaSwag | 27.8 ± 4.8 | 30.2 ± 0.5 | 29.8 ± 0.9 | 29.9 ± 0.7 | 28.5 ± 1.1 | 29.0 ± 0.5 | 27.0 ± 1.2 | 19.7 ± 0.5 |
| PiQA | 50.6 ± 0.5 | 50.8 ± 0.5 | 48.6 ± 3.9 | 50.9 ± 0.7 | 50.3 ± 1.3 | 49.5 ± 0.4 | 47.6 ± 1.2 | 39.5 ± 1.3 |
| RTE | 5.6 ± 3.1 | 2.6 ± 3.9 | 7.2 ± 2.7 | 7.0 ± 3.2 | 8.8 ± 5.3 | 9.3 ± 3.6 | 3.0 ± 4.3 | 2.6 ± 4.2 |
| SciQ | 84.6 ± 3.9 | 86.1 ± 1.3 | 84.4 ± 3.7 | 85.9 ± 0.7 | 86.2 ± 0.8 | 79.0 ± 0.7 | 81.1 ± 1.4 | 65.3 ± 1.1 |
| StoryCloze 2016 | 61.1 ± 3.7 | 62.6 ± 0.2 | 61.9 ± 2.2 | 62.6 ± 0.4 | 61.8 ± 0.8 | 61.5 ± 0.8 | 60.1 ± 0.7 | 53.9 ± 0.5 |
| WinoGrande XL | 17.0 ± 2.6 | 17.8 ± 1.4 | 16.5 ± 1.8 | 17.1 ± 1.8 | 14.9 ± 1.5 | 15.9 ± 1.2 | 11.8 ± 1.5 | 3.9 ± 0.8 |
| E2E NLG | 18.2 ± 1.2 | 18.8 ± 0.8 | 17.8 ± 1.5 | 16.0 ± 2.2 | 15.9 ± 2.5 | 13.8 ± 1.3 | 15.7 ± 0.9 | 11.2 ± 1.4 |
| XSUM | 2.9 ± 0.2 | 3.0 ± 0.2 | 2.8 ± 0.3 | 2.9 ± 0.2 | 2.9 ± 0.2 | 1.0 ± 0.4 | 2.4 ± 0.1 | 1.8 ± 0.1 |
| WebNLG EN | 4.8 ± 2.0 | 5.7 ± 0.2 | 5.4 ± 0.3 | 5.6 ± 0.2 | 5.4 ± 0.5 | 5.5 ± 0.1 | 5.4 ± 0.2 | 3.4 ± 0.3 |
| WikiLingua EN | 3.3 ± 0.5 | 3.6 ± 0.1 | 3.4 ± 0.1 | 3.4 ± 0.1 | 3.3 ± 0.1 | 1.4 ± 0.6 | 3.0 ± 0.1 | 2.2 ± 0.1 |
| bAbI | 0.0 ± 0.0 | 0.0 ± 0.0 | 0.0 ± 0.0 | 0.0 ± 0.0 | 0.0 ± 0.0 | 0.0 ± 0.0 | 0.0 ± 0.0 | 0.0 ± 0.0 |
| Average | 22.1 ± 1.7 | 22.3 ± 0.9 | 21.9 ± 1.2 | 22.8 ± 0.5 | 22.5 ± 0.6 | 21.6 ± 0.5 | 20.6 ± 0.6 | 15.2 ± 1.0 |

Table 7: **Results for 4.2B parameter models trained on repeated data on OSCAR for 84B total tokens.** Scores are normalized averages of 0-5 few-shots and reported as percentages. We report mean/std. err. across five different models, each trained with a different random seed.

| Unique Tokens | 84B | 42B | 28B | 21B | 17B | 12B | 6B | 1.9B |
|---|---|---|---|---|---|---|---|---|
| Epochs | 1 | 2 | 3 | 4 | 5 | 7 | 14 | 44 |
| ANLI R1 | -0.9 ± 0.5 | -0.8 ± 1.1 | -0.9 ± 1.4 | -0.4 ± 0.4 | -0.1 ± 1.2 | 0.3 ± 1.1 | -0.5 ± 0.8 | 1.1 ± 1.3 |
| ANLI R2 | 0.7 ± 0.9 | 0.7 ± 1.1 | 1.3 ± 1.0 | 1.5 ± 1.1 | 1.7 ± 1.3 | 0.9 ± 0.8 | 0.9 ± 1.0 | 1.7 ± 1.3 |
| ANLI R3 | 0.4 ± 0.6 | 0.6 ± 0.4 | 0.7 ± 0.3 | 0.4 ± 0.8 | 0.7 ± 1.2 | 0.6 ± 1.2 | 0.7 ± 0.5 | 0.8 ± 1.2 |
| ARC-Challenge | 1.3 ± 0.5 | 1.8 ± 0.5 | 1.6 ± 0.7 | 2.4 ± 1.1 | 1.6 ± 0.7 | 2.0 ± 0.7 | 1.6 ± 0.5 | -2.1 ± 0.5 |
| ARC-Easy | 45.5 ± 0.8 | 45.1 ± 1.2 | 44.8 ± 0.9 | 44.8 ± 0.6 | 45.0 ± 1.0 | 43.9 ± 0.7 | 40.7 ± 0.7 | 28.0 ± 0.9 |
| BoolQ | 14.5 ± 1.9 | 15.1 ± 4.6 | 10.8 ± 5.1 | 12.5 ± 1.9 | 6.7 ± 4.0 | 10.1 ± 4.2 | -0.0 ± 6.9 | -4.3 ± 7.2 |
| CB | 21.3 ± 2.3 | 19.2 ± 3.8 | 12.9 ± 6.4 | 16.9 ± 3.4 | 15.1 ± 9.4 | 17.8 ± 3.6 | 15.0 ± 8.1 | 11.2 ± 4.1 |
| COPA | 43.1 ± 3.0 | 42.5 ± 3.7 | 44.4 ± 1.1 | 43.0 ± 3.4 | 41.8 ± 2.3 | 44.6 ± 2.7 | 40.3 ± 3.0 | 34.9 ± 4.9 |
| HellaSwag | 21.1 ± 0.2 | 21.0 ± 0.2 | 20.9 ± 0.1 | 20.7 ± 0.2 | 20.5 ± 0.3 | 20.3 ± 0.1 | 19.3 ± 0.1 | 14.5 ± 0.2 |
| PiQA | 45.3 ± 0.9 | 44.8 ± 0.7 | 44.8 ± 0.9 | 44.4 ± 0.6 | 44.3 ± 0.6 | 43.9 ± 0.5 | 42.2 ± 0.9 | 34.0 ± 0.8 |
| RTE | 4.2 ± 2.8 | 1.5 ± 2.4 | -1.1 ± 3.9 | -2.5 ± 3.9 | 5.3 ± 1.8 | 4.4 ± 1.9 | 1.6 ± 2.2 | -1.0 ± 2.4 |
| SciQ | 86.6 ± 0.7 | 86.5 ± 0.5 | 86.0 ± 0.2 | 86.3 ± 1.0 | 85.4 ± 0.8 | 84.7 ± 0.4 | 82.0 ± 1.4 | 62.9 ± 2.5 |
| StoryCloze 2016 | 56.5 ± 0.6 | 56.8 ± 0.6 | 56.5 ± 0.7 | 55.8 ± 0.3 | 55.9 ± 0.2 | 56.0 ± 0.3 | 54.5 ± 0.7 | 49.3 ± 0.2 |
| WinoGrande XL | 9.7 ± 1.4 | 9.0 ± 1.8 | 9.5 ± 0.7 | 8.9 ± 1.0 | 7.8 ± 1.2 | 7.4 ± 1.4 | 6.8 ± 1.4 | 2.1 ± 1.0 |
| E2E NLG | 21.4 ± 1.3 | 21.9 ± 0.4 | 21.2 ± 1.0 | 21.8 ± 0.6 | 21.0 ± 0.9 | 20.5 ± 0.7 | 20.9 ± 1.0 | 16.0 ± 0.6 |
| XSUM | 3.6 ± 0.2 | 3.5 ± 0.2 | 3.5 ± 0.2 | 3.5 ± 0.2 | 3.5 ± 0.3 | 3.2 ± 0.5 | 3.0 ± 0.2 | 1.9 ± 0.1 |
| WebNLG EN | 9.9 ± 0.4 | 9.7 ± 0.8 | 9.3 ± 0.6 | 9.7 ± 0.5 | 9.3 ± 0.7 | 9.4 ± 0.3 | 8.9 ± 0.5 | 3.8 ± 0.4 |
| WikiLingua EN | 3.9 ± 0.1 | 3.8 ± 0.2 | 3.6 ± 0.3 | 3.7 ± 0.2 | 3.6 ± 0.2 | 3.7 ± 0.1 | 3.3 ± 0.2 | 2.1 ± 0.2 |
| bAbI | 15.0 ± 7.5 | 19.0 ± 1.2 | 18.8 ± 1.4 | 18.5 ± 1.4 | 19.2 ± 0.6 | 18.1 ± 1.4 | 14.5 ± 1.5 | 9.6 ± 1.7 |
| Average | 21.2 ± 0.2 | 21.1 ± 0.4 | 20.4 ± 0.3 | 20.6 ± 0.5 | 20.4 ± 0.5 | 20.6 ± 0.2 | 18.7 ± 0.5 | 14.0 ± 0.6 |

Table 8: **Results for 8.7B parameter models trained on repeated data on C4 for 178B total tokens and a data-constrained compute-optimal 6.3B model.** Scores are normalized averages of 0-5 few-shots and reported as percentages. The two models with 25 billion unique tokens are the ones depicted in Figure 1 (right). The data-constrained compute-optimal variant (6.3 billion parameters) performs better by using fewer parameters and repeating more data.

| Parameters | 8.7B | | | | | | | | 6.3B |
|---|---|---|---|---|---|---|---|---|---|
| Unique Tokens | 178B | 88B | 58B | 44B | 35B | 25B | 13B | 4B | 25B |
| Epochs | 1 | 2 | 3 | 4 | 5 | 7 | 14 | 44 | 9.7 |
| ANLI R1 | -0.9 | -1.2 | -4.2 | 0.7 | -1.3 | 0.1 | 1.2 | 2.1 | -0.9 |
| ANLI R2 | -0.4 | -1.2 | -0.2 | 0.2 | -0.4 | -0.1 | 0.4 | 2.2 | 1.0 |
| ANLI R3 | 0.7 | 0.5 | 0.7 | 1.8 | 0.4 | 1.6 | 2.0 | 4.0 | 2.6 |
| ARC-Challenge | 12.2 | 11.9 | 10.5 | 12.2 | 10.6 | 11.8 | 8.3 | 2.2 | 12.7 |
| ARC-Easy | 58.5 | 58.0 | 56.9 | 57.4 | 56.7 | 58.5 | 52.9 | 37.4 | 57.2 |
| BoolQ | 26.1 | 31.8 | 31.3 | 30.3 | 28.8 | 28.5 | 27.9 | 4.1 | 30.6 |
| CB | 7.6 | 12.9 | -15.2 | 17.9 | 14.3 | -22.8 | -12.1 | 17.4 | 6.2 |
| COPA | 68.0 | 64.7 | 62.3 | 66.3 | 63.3 | 70.0 | 57.0 | 45.0 | 66.0 |
| HellaSwag | 37.8 | 37.8 | 37.3 | 37.4 | 37.1 | 37.5 | 36.1 | 27.5 | 38.1 |
| PiQA | 55.9 | 55.6 | 54.7 | 56.5 | 55.8 | 53.9 | 52.4 | 45.7 | 54.3 |
| RTE | 14.1 | 11.4 | 11.0 | 8.7 | 15.9 | -2.6 | -1.8 | -3.2 | 7.7 |
| SciQ | 90.4 | 91.1 | 90.7 | 90.0 | 89.8 | 89.8 | 87.9 | 72.9 | 90.3 |
| StoryCloze 2016 | 68.3 | 67.3 | 67.2 | 67.6 | 67.8 | 66.8 | 66.2 | 58.9 | 68.4 |
| WinoGrande XL | 26.3 | 27.7 | 26.5 | 29.0 | 26.1 | 23.5 | 18.1 | 10.0 | 27.0 |
| E2E NLG | 20.5 | 17.9 | 18.7 | 20.0 | 17.2 | 17.7 | 17.4 | 11.2 | 16.9 |
| XSUM | 3.6 | 3.3 | 3.8 | 3.8 | 3.5 | 3.0 | 3.3 | 2.0 | 3.8 |
| WebNLG EN | 5.3 | 5.8 | 5.9 | 5.6 | 5.8 | 5.2 | 5.7 | 4.9 | 5.3 |
| WikiLingua EN | 4.1 | 4.2 | 4.2 | 4.1 | 4.2 | 4.0 | 3.5 | 2.7 | 4.0 |
| bAbI | 0.0 | 0.0 | 0.0 | 0.0 | 0.0 | 0.0 | 0.0 | 0.0 | 0.0 |
| Average | 26.2 | 26.3 | 24.3 | 26.8 | 26.1 | 23.5 | 22.4 | 18.3 | 25.9 |

Table 9: **Results for 8.7B parameter models trained on repeated data on OSCAR for 178B total tokens.** Scores are normalized averages of 0-5 few-shots and reported as percentages.

| Unique Tokens | 178B | 88B | 58B | 44B | 35B | 25B | 13B | 4B |
|---|---|---|---|---|---|---|---|---|
| Epochs | 1 | 2 | 3 | 4 | 5 | 7 | 14 | 44 |
| ANLI R1 | -1.3 | -2.3 | -0.5 | -1.8 | 0.1 | -0.3 | 2.6 | -0.4 |
| ANLI R2 | 0.8 | 3.2 | -0.2 | -1.3 | 1.0 | 0.2 | 1.5 | 0.5 |
| ANLI R3 | 1.1 | 1.2 | 1.3 | 0.9 | 2.8 | -0.4 | 1.1 | -0.1 |
| ARC-Challenge | 6.9 | 6.7 | 6.9 | 3.8 | 6.6 | 4.8 | 4.0 | -0.9 |
| ARC-Easy | 50.2 | 51.6 | 51.2 | 51.0 | 51.9 | 50.8 | 47.0 | 33.0 |
| BoolQ | 18.4 | 11.7 | 19.4 | 22.4 | 17.5 | 20.8 | 7.6 | 4.1 |
| CB | 11.2 | 13.4 | 16.1 | 19.6 | 21.4 | 25.0 | 9.8 | 20.1 |
| COPA | 46.7 | 53.0 | 52.0 | 53.7 | 51.0 | 53.3 | 48.7 | 41.7 |
| HellaSwag | 27.4 | 27.2 | 26.8 | 26.8 | 27.3 | 26.7 | 25.5 | 19.6 |
| PiQA | 49.2 | 49.3 | 50.1 | 48.7 | 48.1 | 47.2 | 45.6 | 37.0 |
| RTE | -0.5 | 1.1 | 0.2 | 1.2 | 10.2 | 3.2 | -3.0 | -7.8 |
| SciQ | 88.1 | 88.0 | 88.4 | 87.9 | 87.9 | 87.4 | 86.3 | 64.6 |
| StoryCloze 2016 | 61.6 | 61.1 | 60.2 | 60.6 | 61.3 | 59.0 | 58.8 | 52.7 |
| WinoGrande XL | 17.6 | 16.3 | 15.4 | 13.7 | 13.9 | 12.8 | 10.8 | -0.6 |
| E2E NLG | 23.3 | 24.2 | 22.2 | 22.9 | 23.1 | 22.1 | 22.9 | 16.8 |
| XSUM | 4.2 | 3.8 | 3.9 | 3.8 | 4.3 | 4.0 | 3.2 | 2.4 |
| WebNLG EN | 9.9 | 10.1 | 10.0 | 10.5 | 9.5 | 9.9 | 10.7 | 5.2 |
| WikiLingua EN | 4.3 | 4.0 | 4.1 | 3.7 | 4.3 | 4.2 | 4.0 | 2.7 |
| bAbI | 20.4 | 20.6 | 21.7 | 21.4 | 21.1 | 21.3 | 19.4 | 10.7 |
| Average | 23.1 | 23.4 | 23.6 | 23.7 | 24.4 | 23.8 | 21.4 | 15.9 |

## M  Detailed Code Augmentation Results

We report tabular results for replacing part of C4 or OSCAR with code for 4.2 billion parameter and 2.8 billion parameter models in Tables 10-11. We find that training on up to 50% of Python data maintains performance on all natural language tasks while enabling huge performance gains on state-tracking (bAbI) for C4. For OSCAR gains are less clear, which is likely due to OSCAR containing code [83], while code data was explicitly filtered out for C4 [90].

Table 10: **Results for code-augmentation for 4.2B parameter models.** Models trained on a mix of natural language (C4) and Python (The Stack). Scores are normalized averages of 0-5 few-shots and reported as percentages. We report mean/std. err. across five different models, each trained with a different random seed.

| | % of Python pre-training data (remainder is C4) | | | | | | | | | |
|---|---|---|---|---|---|---|---|---|---|---|
| Dataset (↓) | 0 | 10 | 20 | 30 | 40 | 50 | 60 | 70 | 80 | 90 |
| ANLI R1 | -1.0 ± 0.3 | -0.7 ± 0.6 | 0.5 ± 0.6 | -0.2 ± 1.1 | -1.5 ± 0.7 | -0.7 ± 1.1 | -1.1 ± 0.9 | -1.2 ± 1.1 | -1.4 ± 0.7 | -1.4 ± 0.6 |
| ANLI R2 | 0.8 ± 0.5 | 0.4 ± 0.8 | 0.3 ± 1.0 | 0.6 ± 0.5 | 0.5 ± 0.6 | 0.3 ± 0.6 | 0.8 ± 0.7 | 0.4 ± 0.5 | 0.1 ± 1.2 | 1.0 ± 0.3 |
| ANLI R3 | 1.1 ± 0.7 | 0.6 ± 0.5 | 0.5 ± 0.6 | 0.2 ± 0.4 | 0.3 ± 0.5 | 0.2 ± 0.5 | 0.3 ± 0.2 | -0.1 ± 0.3 | -0.0 ± 0.2 | -0.1 ± 0.2 |
| ARC-Challenge | 5.3 ± 0.6 | 6.4 ± 1.0 | 5.2 ± 2.4 | 4.3 ± 1.4 | 5.2 ± 0.8 | 5.2 ± 0.5 | 2.6 ± 0.4 | 1.7 ± 0.5 | -0.4 ± 0.4 | -3.0 ± 0.4 |
| ARC-Easy | 49.2 ± 0.9 | 52.4 ± 1.1 | 49.6 ± 3.7 | 48.1 ± 4.1 | 50.1 ± 1.0 | 49.7 ± 0.3 | 48.0 ± 0.5 | 45.6 ± 0.5 | 43.3 ± 0.4 | 37.7 ± 0.7 |
| BoolQ | 18.2 ± 4.0 | 10.5 ± 12.0 | 16.3 ± 5.2 | 17.8 ± 3.3 | 13.4 ± 3.4 | 14.8 ± 2.1 | 12.5 ± 8.9 | 12.1 ± 6.6 | 7.2 ± 6.7 | 10.7 ± 7.3 |
| CB | 12.0 ± 7.2 | 20.3 ± 2.7 | 14.4 ± 7.1 | 16.5 ± 1.2 | 22.3 ± 3.1 | 22.1 ± 4.8 | 19.4 ± 4.8 | 23.8 ± 3.3 | 23.8 ± 4.1 | 23.4 ± 2.4 |
| COPA | 59.1 ± 5.4 | 56.4 ± 4.9 | 46.7 ± 8.7 | 50.2 ± 3.7 | 52.7 ± 2.5 | 50.1 ± 4.5 | 46.5 ± 1.9 | 43.1 ± 4.2 | 39.2 ± 3.7 | 35.9 ± 4.9 |
| HellaSwag | 27.8 ± 4.8 | 29.4 ± 0.4 | 25.7 ± 4.8 | 27.0 ± 1.7 | 26.3 ± 2.4 | 26.3 ± 0.6 | 25.0 ± 0.1 | 22.6 ± 0.1 | 19.5 ± 0.2 | 14.7 ± 0.1 |
| PiQA | 50.6 ± 0.5 | 50.8 ± 0.6 | 48.6 ± 3.0 | 48.2 ± 2.8 | 48.7 ± 0.7 | 48.4 ± 1.0 | 47.1 ± 0.7 | 45.6 ± 0.4 | 43.4 ± 0.9 | 39.0 ± 0.8 |
| RTE | 5.6 ± 3.1 | 7.3 ± 3.4 | 4.4 ± 4.7 | 6.1 ± 2.6 | 9.1 ± 4.0 | 8.1 ± 5.9 | 7.7 ± 5.3 | 4.0 ± 2.1 | 6.2 ± 2.1 | 4.6 ± 2.5 |
| SciQ | 84.6 ± 3.9 | 87.1 ± 0.2 | 84.6 ± 4.8 | 86.9 ± 1.2 | 86.9 ± 1.2 | 87.9 ± 0.9 | 87.6 ± 0.6 | 87.0 ± 0.2 | 86.0 ± 0.2 | 84.5 ± 0.6 |
| StoryCloze 2016 | 61.1 ± 3.7 | 62.0 ± 0.6 | 59.0 ± 4.8 | 60.8 ± 1.5 | 59.9 ± 1.9 | 60.0 ± 0.7 | 59.0 ± 0.4 | 57.2 ± 0.5 | 54.9 ± 0.4 | 51.0 ± 0.3 |
| WinoGrande XL | 17.0 ± 2.6 | 17.4 ± 2.1 | 14.9 ± 4.4 | 15.2 ± 2.0 | 15.7 ± 1.2 | 14.2 ± 1.0 | 13.5 ± 1.3 | 10.7 ± 1.3 | 9.1 ± 0.6 | 5.3 ± 1.3 |
| E2E NLG | 18.2 ± 1.2 | 21.8 ± 1.6 | 15.9 ± 8.6 | 23.3 ± 0.6 | 21.5 ± 3.8 | 23.9 ± 0.6 | 23.7 ± 0.6 | 23.7 ± 0.5 | 24.3 ± 0.7 | 24.0 ± 0.9 |
| XSUM | 2.9 ± 0.2 | 3.2 ± 0.5 | 3.4 ± 0.3 | 3.3 ± 0.3 | 3.6 ± 0.6 | 3.4 ± 0.2 | 3.5 ± 0.2 | 2.9 ± 0.3 | 2.8 ± 0.4 | 2.7 ± 0.2 |
| WebNLG EN | 4.8 ± 2.0 | 9.5 ± 0.7 | 10.2 ± 1.1 | 10.5 ± 0.7 | 10.4 ± 0.8 | 10.4 ± 0.6 | 9.9 ± 0.4 | 10.0 ± 0.5 | 9.3 ± 0.6 | 9.2 ± 0.2 |
| WikiLingua EN | 3.3 ± 0.5 | 4.0 ± 0.1 | 4.0 ± 0.2 | 4.2 ± 0.1 | 4.3 ± 0.3 | 4.2 ± 0.2 | 4.4 ± 0.3 | 4.1 ± 0.2 | 3.9 ± 0.2 | 3.6 ± 0.3 |
| bAbI | 0.0 ± 0.0 | 12.5 ± 6.7 | 13.8 ± 7.2 | 15.8 ± 8.2 | 17.4 ± 9.2 | 23.2 ± 1.2 | 23.4 ± 2.0 | 24.3 ± 1.4 | 23.2 ± 1.0 | 24.6 ± 1.8 |
| Average | 22.1 ± 1.7 | 23.7 ± 0.7 | 22.0 ± 3.0 | 23.1 ± 1.1 | 23.5 ± 1.0 | 23.8 ± 0.5 | 22.8 ± 1.0 | 22.0 ± 0.6 | 20.8 ± 0.5 | 19.3 ± 0.3 |
| Average (no bAbI) | 23.4 ± 1.8 | 24.4 ± 0.7 | 22.5 ± 2.8 | 23.5 ± 0.8 | 23.9 ± 0.9 | 23.8 ± 0.5 | 22.8 ± 1.0 | 21.8 ± 0.6 | 20.6 ± 0.6 | 19.1 ± 0.4 |

Table 11: **Results for code-augmentation for 2.8B parameter models.** Models trained on a mix of natural language (C4) and Python (The Stack). Scores are normalized averages of 0-5 few-shots and reported as percentages. We report mean/std. err. across five different models, each trained with a different random seed.

| Dataset (↓) | % of Python pre-training data (rest is C4) | | | | | | % of Python pre-training data (rest is OSCAR) | | | | | |
| --- | --- | --- | --- | --- | --- | --- | --- | --- | --- | --- | --- | --- |
| | 0 | 10 | 20 | 30 | 40 | 50 | 0 | 10 | 20 | 30 | 40 | 50 |
| ANLI R1 | 0.4 ± 1.6 | -1.5 | -0.9 | -1.0 | -0.7 | -2.4 | -0.3 ± 0.5 | 0.0 | -0.6 | -1.6 | -2.4 | -1.7 |
| ANLI R2 | 0.9 ± 0.4 | 0.7 | 0.0 | 0.1 | -0.1 | 0.1 | 1.0 ± 1.0 | 1.2 | -0.1 | -0.0 | 0.0 | 0.8 |
| ANLI R3 | 1.7 ± 0.5 | 0.6 | -0.7 | -0.2 | 0.4 | 0.0 | 0.4 ± 0.8 | -0.4 | -0.2 | -1.7 | -0.8 | -0.5 |
| ARC-Challenge | 1.6 ± 1.0 | 4.2 | 1.7 | 1.5 | 0.2 | -0.2 | -1.4 ± 0.8 | -0.7 | -1.4 | -3.4 | -2.3 | -3.1 |
| ARC-Easy | 44.5 ± 0.5 | 46.4 | 46.5 | 45.4 | 43.6 | 42.7 | 39.7 ± 0.3 | 39.8 | 38.7 | 39.1 | 37.3 | 37.6 |
| BoolQ | 18.8 ± 3.4 | 15.7 | 19.0 | 13.4 | 16.0 | 4.4 | 12.8 ± 4.4 | 3.3 | 12.5 | 10.6 | 5.8 | 8.5 |
| CB | 20.0 ± 4.7 | 22.8 | 10.7 | 20.5 | 17.4 | 15.2 | 19.7 ± 5.1 | 14.7 | 15.6 | 19.6 | 22.8 | 17.0 |
| COPA | 49.7 ± 3.5 | 46.3 | 49.3 | 46.3 | 42.7 | 40.0 | 42.7 ± 2.2 | 42.7 | 41.0 | 42.7 | 35.7 | 38.0 |
| HellaSwag | 24.7 ± 0.3 | 24.1 | 23.3 | 22.3 | 21.9 | 20.9 | 16.3 ± 0.1 | 15.7 | 15.9 | 15.5 | 15.1 | 13.7 |
| PiQA | 47.9 ± 0.6 | 46.9 | 47.7 | 45.1 | 46.2 | 45.5 | 41.2 ± 0.7 | 41.6 | 39.9 | 40.5 | 38.8 | 38.6 |
| RTE | 5.1 ± 4.0 | 8.8 | 7.7 | 5.1 | 7.8 | 10.8 | 3.9 ± 1.1 | 2.2 | 4.3 | 1.1 | 3.7 | -1.7 |
| SciQ | 83.2 ± 0.6 | 83.3 | 85.3 | 84.8 | 83.2 | 83.7 | 83.2 ± 0.6 | 82.4 | 83.5 | 82.8 | 83.3 | 83.2 |
| StoryCloze 2016 | 58.7 ± 0.2 | 59.3 | 57.9 | 56.9 | 56.5 | 56.0 | 52.8 ± 0.3 | 52.0 | 52.2 | 52.0 | 51.8 | 50.9 |
| WinoGrande XL | 11.6 ± 0.8 | 13.0 | 10.7 | 9.3 | 8.2 | 9.6 | 5.8 ± 0.9 | 3.2 | 5.6 | 5.8 | 4.6 | 3.9 |
| E2E NLG | 17.0 ± 1.4 | 19.8 | 21.1 | 20.2 | 22.1 | 21.0 | 20.3 ± 0.3 | 21.9 | 20.7 | 20.5 | 20.7 | 21.1 |
| XSUM | 2.4 ± 0.1 | 2.7 | 2.0 | 2.2 | 2.0 | 2.3 | 3.0 ± 0.1 | 2.8 | 3.1 | 3.4 | 3.1 | 2.9 |
| WebNLG EN | 5.3 ± 0.1 | 9.1 | 8.0 | 8.5 | 8.5 | 9.1 | 8.8 ± 0.4 | 8.7 | 9.6 | 9.1 | 8.7 | 9.4 |
| WikiLingua EN | 3.0 ± 0.1 | 3.2 | 3.2 | 3.6 | 3.3 | 3.7 | 2.9 ± 0.1 | 3.3 | 3.6 | 3.5 | 3.4 | 3.5 |
| bAbI | 0.0 ± 0.0 | 4.6 | 14.2 | 14.2 | 14.8 | 15.1 | 15.5 ± 1.0 | 16.6 | 17.2 | 17.2 | 17.7 | 15.9 |
| Average | 20.9 ± 0.4 | 21.6 | 21.4 | 21.0 | 20.7 | 19.9 | 19.4 ± 0.5 | 18.5 | 19.0 | 18.8 | 18.3 | 17.8 |
| Average (without bAbI) | 22.0 ± 0.5 | 22.5 | 21.8 | 21.3 | 21.1 | 20.1 | 19.6 ± 0.5 | 18.6 | 19.1 | 18.9 | 18.3 | 17.9 |

# N    Filtering Procedure

**Perplexity filtering**    We follow the approach of [54] to perform perplexity filtering and reuse their artifacts - a SentencePiece tokenizer [52] and a KenLM 5-gram language model [37] trained on Wikipedia introductions and available to download from their repository.[4] We compute the model's perplexity on all OSCAR and C4 samples and only select samples that fall within a certain percentile threshold. For example, to select the top 25%, we only select samples with perplexity lower than the 25th percentile. Figure 18 provides a visual representation of perplexity distribution for respective datasets, highlighting the relevant percentile thresholds.

**Deduplication**    We perform deduplication leveraging the suffix array-based approach proposed by Lee et al. [55]. We remove any document with at least a 100-character span overlapping with any other document in the corpus. We deduplicate the full C4 dataset. In the case of OSCAR, the memory requirements of the deduplication procedure make performing the full dataset deduplication infeasible. Instead, we select a 25% subset of the full OSCAR and build a suffix array for this subset. We experiment with leveraging the 25% OSCAR suffix array in two ways. First, we deduplicate the selected subset. This is very strict and preserves less than 5% of the full OSCAR. Subsequently, we use the 25% suffix array to deduplicate the full OSCAR, i.e. we remove any document which has at least a 100-character span overlapping with the 25% subset we selected. This is more permissive and allows us to preserve 31% of the original dataset. We refer to the latter as *expanded* in Table 12 and it is used for the training of the 4.2 billion parameter model in Table 14, while the smaller deduplicated version of OSCAR is used for the 2.8 billion parameter model.

**ROOTS filter**    In addition, we benchmark with the filtering procedure from the ROOTS corpus [54]. It applies the following set of filters:

- Discarding documents with too few words
- Discarding documents with overly repeated character- and word-n-grams
- Discarding documents with too many special characters

---

[4]https://github.com/bigscience-workshop/data-preparation/tree/main/preprocessing/training/01b_oscar_cleaning_and_filtering

- Discarding documents with too few grammatical function words (e.g. "of", "and")
- Discarding documents with too many flagged words
- Discarding documents with a low `fasttext` language identification score
- Perplexity filtering

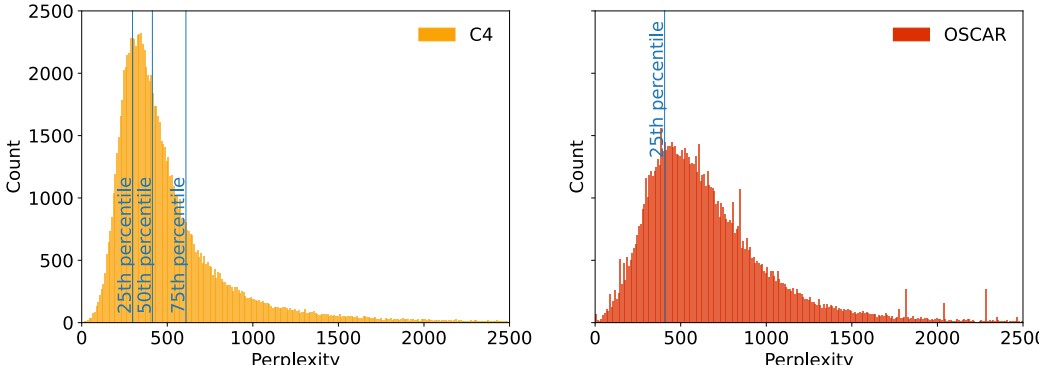

Figure 18: Perplexity histograms for respective datasets. For demonstration purposes, we use 100,000 random samples of each dataset.

Table 12: **Sizes of filtered datasets.**

| Base Dataset | Filter | Tokens after filtering |
|---|---|---|
| C4 | Deduplication | 21 billion |
| C4 | Perplexity Top 25% | 44 billion |
| C4 | Perplexity Top 50% | 89 billion |
| C4 | Perplexity 25-75% | 89 billion |
| OSCAR | Deduplication | 9 billion |
| OSCAR | Deduplication-expanded | 94 billion |
| OSCAR | Perplexity Top 25% | 80 billion |
| OSCAR | ROOTS | 99 billion |

## O   Detailed Filtering Results

In Table 13, we report detailed perplexity filtering results on C4 and OSCAR. For C4, perplexity filtering is only effective at 4.2B parameters. Meanwhile, for OSCAR, which is noisier than C4, perplexity filtering seems effective both for 2.8B and 4.2B parameters. Table 14 contains deduplication results and results for the ROOTS filter. Deduplication does not improve downstream performance for C4 while being effective for OSCAR which has significantly more noise. Applying the ROOTS filter on OSCAR is not better than the unfiltered OSCAR on our benchmark, but might have other beneficial effects, such as reducing obscenity, templated messages, or repetition, depending on the final use case.

Table 13: **Results for perplexity-filtering.** The training data is perplexity filtered according to the given percentile, e.g. 25% corresponds to the top 25% percent of examples with the lowest perplexity. The resulting dataset sizes are in Table 12. The data is repeated until it matches 55B tokens for 2.8B parameter and 84B tokens for 4.2B parameter models. Scores are normalized averages of 0-5 few-shots and reported as percentages. For unfiltered models we report mean/std. err. across five different models, each trained with a different random seed.

| Training Data | C4 | | | | | | | OSCAR | | | |
|---|---|---|---|---|---|---|---|---|---|---|---|
| Parameters | 2.8B | | | 4.2B | | | | 2.8B | | 4.2B | |
| Percentile | All | 25% | 50% | All | 25% | 50% | 25-75% | All | 25% | All | 25% |
| ANLI R1 | 0.4 ± 1.6 | -0.1 | 0.9 | -0.5 ± 1.4 | -0.0 | -0.7 | -0.8 | -0.3 ± 0.5 | -0.4 | -0.4 ± 1.2 | -2.2 |
| ANLI R2 | 0.9 ± 0.4 | -0.2 | -0.7 | 0.0 ± 1.3 | -0.4 | -0.0 | 1.1 | 1.0 ± 1.0 | 1.7 | 1.0 ± 0.9 | 0.7 |
| ANLI R3 | 1.7 ± 0.5 | 0.5 | 1.4 | 0.7 ± 0.5 | 0.7 | 2.9 | 0.4 | 0.4 ± 0.8 | 1.7 | 1.2 ± 0.5 | 2.1 |
| ARC-Challenge | 1.6 ± 1.0 | 3.3 | 2.9 | 4.2 ± 1.6 | 10.2 | 9.3 | 7.9 | -1.4 ± 0.8 | 3.3 | 1.8 ± 0.8 | 6.3 |
| ARC-Easy | 44.5 ± 0.5 | 47.3 | 47.7 | 48.1 ± 4.8 | 55.8 | 53.7 | 51.0 | 39.7 ± 0.3 | 46.8 | 45.7 ± 0.6 | 51.8 |
| BoolQ | 18.8 ± 3.4 | 17.1 | 17.7 | 22.4 ± 3.3 | 27.7 | 23.5 | 24.5 | 12.8 ± 4.4 | 11.8 | 12.4 ± 5.9 | 22.2 |
| CB | 20.0 ± 4.7 | 16.1 | 13.8 | 9.3 ± 16.6 | 24.6 | 22.3 | 12.5 | 19.7 ± 5.1 | 17.0 | 23.9 ± 3.8 | 20.1 |
| COPA | 49.7 ± 3.5 | 55.7 | 56.0 | 55.3 ± 3.8 | 60.7 | 66.0 | 61.0 | 42.7 ± 2.2 | 44.0 | 41.1 ± 3.0 | 49.3 |
| HellaSwag | 24.7 ± 0.3 | 24.7 | 26.0 | 29.4 ± 1.3 | 30.7 | 32.7 | 33.1 | 16.3 ± 0.1 | 19.0 | 21.0 ± 0.2 | 23.3 |
| PiQA | 47.9 ± 0.6 | 43.4 | 45.8 | 48.8 ± 3.8 | 47.9 | 52.2 | 52.1 | 41.2 ± 0.7 | 38.3 | 45.0 ± 0.6 | 44.4 |
| RTE | 5.1 ± 4.0 | 5.7 | 7.3 | 6.9 ± 3.1 | 11.9 | 2.2 | 10.3 | 3.9 ± 1.1 | -1.2 | 2.2 ± 4.3 | 7.0 |
| SciQ | 83.2 ± 0.6 | 82.4 | 82.8 | 86.3 ± 1.1 | 88.6 | 87.4 | 88.4 | 83.2 ± 0.6 | 84.0 | 86.3 ± 0.6 | 86.5 |
| StoryCloze 2016 | 58.7 ± 0.2 | 61.1 | 61.2 | 62.8 ± 0.5 | 65.5 | 65.6 | 65.1 | 52.8 ± 0.3 | 57.9 | 57.2 ± 0.6 | 60.2 |
| WinoGrande XL | 11.6 ± 0.8 | 15.3 | 14.3 | 18.7 ± 1.0 | 24.9 | 22.3 | 18.7 | 5.8 ± 0.9 | 9.7 | 10.1 ± 1.0 | 14.8 |
| E2E NLG | 17.0 ± 1.4 | 16.1 | 16.8 | 17.9 ± 0.7 | 18.8 | 17.8 | 19.2 | 20.3 ± 0.3 | 19.5 | 21.6 ± 0.7 | 22.6 |
| XSUM | 2.4 ± 0.1 | 2.6 | 3.0 | 3.0 ± 0.3 | 3.9 | 3.2 | 3.0 | 3.0 ± 0.1 | 3.2 | 3.7 ± 0.2 | 2.7 |
| WebNLG EN | 5.3 ± 0.1 | 4.8 | 5.1 | 5.6 ± 0.3 | 5.4 | 5.7 | 5.2 | 8.8 ± 0.4 | 6.9 | 9.3 ± 0.5 | 10.6 |
| WikiLingua EN | 3.0 ± 0.1 | 3.2 | 3.3 | 3.6 ± 0.2 | 3.4 | 3.5 | 3.4 | 2.9 ± 0.1 | 3.4 | 4.0 ± 0.1 | 3.8 |
| bAbI | 0.0 ± 0.0 | 0.0 | 0.0 | 0.0 ± 0.0 | 0.0 | 0.0 | 0.0 | 15.5 ± 1.0 | 14.5 | 19.3 ± 1.0 | 17.2 |
| Average | 20.9 ± 0.4 | 21.0 | 21.3 | 22.2 ± 1.4 | 25.3 | 24.7 | 24.0 | 19.4 ± 0.5 | 20.1 | 21.4 ± 0.5 | 23.3 |

Table 14: **Results for filtering with deduplication and the ROOTS filters.** The resulting dataset sizes are in Table 12. The data is repeated until it matches 55B tokens for 2.8B parameter and 84B tokens for 4.2B parameter models. Scores are normalized averages of 0-5 few-shots and reported as percentages. For unfiltered models we report mean/std. err. across five different models, each trained with a different random seed.

| Training Data | C4 | | | | OSCAR | | | | | |
|---|---|---|---|---|---|---|---|---|---|---|
| Parameters | 2.8B parameters | | 4.2B parameters | | 2.8B parameters | | | 4.2B parameters | | |
| Method | All | Dedup. | All | Dedup. | All | Dedup. | ROOTS | All | Dedup.-exp. | ROOTS |
| ANLI R1 | 0.4 ± 1.6 | -0.2 | -0.5 ± 1.4 | -0.8 | -0.3 ± 0.5 | -2.1 | -1.7 | -0.4 ± 1.2 | -1.8 | 1.2 |
| ANLI R2 | 0.9 ± 0.4 | 1.1 | 0.0 ± 1.3 | -0.1 | 1.0 ± 1.0 | 2.0 | 0.7 | 1.0 ± 0.9 | -0.5 | -0.3 |
| ANLI R3 | 1.7 ± 0.5 | 1.8 | 0.7 ± 0.5 | 0.4 | 0.4 ± 0.8 | 0.4 | 0.2 | 1.2 ± 0.5 | 0.8 | -0.3 |
| ARC-Challenge | 1.6 ± 1.0 | 0.6 | 4.2 ± 1.6 | 3.9 | -1.4 ± 0.8 | 2.6 | -0.9 | 1.8 ± 0.8 | 6.8 | 0.6 |
| ARC-Easy | 44.5 ± 0.5 | 43.0 | 48.1 ± 4.8 | 46.8 | 39.7 ± 0.3 | 44.6 | 42.3 | 45.7 ± 0.6 | 51.0 | 47.1 |
| BoolQ | 18.8 ± 3.4 | 1.5 | 22.4 ± 3.3 | 2.2 | 12.8 ± 4.4 | 3.4 | 13.4 | 12.4 ± 5.9 | 13.0 | 7.0 |
| CB | 20.0 ± 4.7 | 0.4 | 9.3 ± 16.6 | 0.9 | 19.7 ± 5.1 | 25.4 | 14.3 | 23.9 ± 3.8 | 25.0 | 28.1 |
| COPA | 49.7 ± 3.5 | 57.0 | 55.3 ± 3.8 | 60.0 | 42.7 ± 2.2 | 47.3 | 37.7 | 41.1 ± 3.0 | 55.3 | 43.0 |
| HellaSwag | 24.7 ± 0.3 | 25.1 | 29.4 ± 1.3 | 30.7 | 16.3 ± 0.1 | 22.8 | 17.6 | 21.0 ± 0.2 | 26.3 | 22.4 |
| PiQA | 47.9 ± 0.6 | 49.1 | 48.8 ± 3.8 | 53.4 | 41.2 ± 0.7 | 45.1 | 41.9 | 45.0 ± 0.6 | 48.5 | 46.3 |
| RTE | 5.1 ± 4.0 | 3.2 | 6.9 ± 3.1 | 0.1 | 3.9 ± 1.1 | 6.1 | 5.8 | 2.2 ± 4.3 | 1.1 | 8.9 |
| SciQ | 83.2 ± 0.6 | 80.4 | 86.3 ± 1.1 | 82.2 | 83.2 ± 0.6 | 82.6 | 83.1 | 86.3 ± 0.6 | 88.5 | 86.4 |
| StoryCloze 2016 | 58.7 ± 0.2 | 61.8 | 62.8 ± 0.5 | 65.2 | 52.8 ± 0.3 | 58.1 | 54.3 | 57.2 ± 0.6 | 61.6 | 58.6 |
| WinoGrande XL | 11.6 ± 0.8 | 13.3 | 18.7 ± 1.0 | 19.7 | 5.8 ± 0.9 | 12.7 | 5.6 | 10.1 ± 1.0 | 16.2 | 11.0 |
| E2E NLG | 17.0 ± 1.4 | 15.6 | 17.9 ± 0.7 | 14.2 | 20.3 ± 0.3 | 20.5 | 20.5 | 21.6 ± 0.7 | 2.4 | 22.6 |
| XSUM | 2.4 ± 0.1 | 2.1 | 3.0 ± 0.3 | 2.5 | 3.0 ± 0.1 | 3.2 | 3.1 | 3.7 ± 0.2 | 4.6 | 3.8 |
| WebNLG EN | 5.3 ± 0.1 | 4.3 | 5.6 ± 0.3 | 4.4 | 8.8 ± 0.4 | 7.4 | 7.4 | 9.3 ± 0.5 | 9.7 | 9.4 |
| WikiLingua EN | 3.0 ± 0.1 | 3.2 | 3.6 ± 0.2 | 3.2 | 2.9 ± 0.1 | 3.0 | 3.1 | 4.0 ± 0.1 | 4.3 | 4.0 |
| bAbI | 0.0 ± 0.0 | 0.0 | 0.0 ± 0.0 | 0.0 | 15.5 ± 1.0 | 17.2 | 14.3 | 19.3 ± 1.0 | 21.1 | 18.0 |
| Average | 20.9 ± 0.4 | 19.1 | 22.2 ± 1.4 | 20.5 | 19.4 ± 0.5 | 21.2 | 19.1 | 21.4 ± 0.5 | 22.8 | 22.0 |

# P  Loss Curves for Complementary Strategies

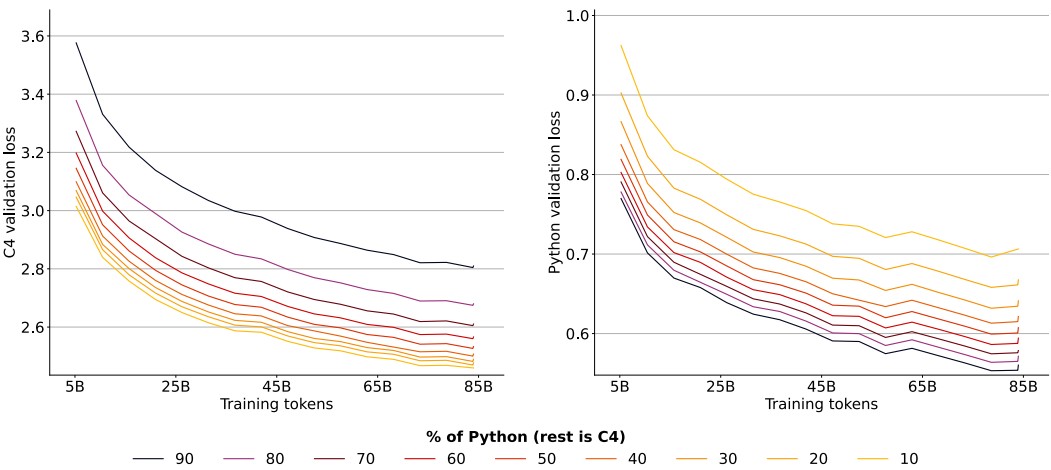

Figure 19: **Validation loss of models trained on a mix of natural language (C4) and Python data.**

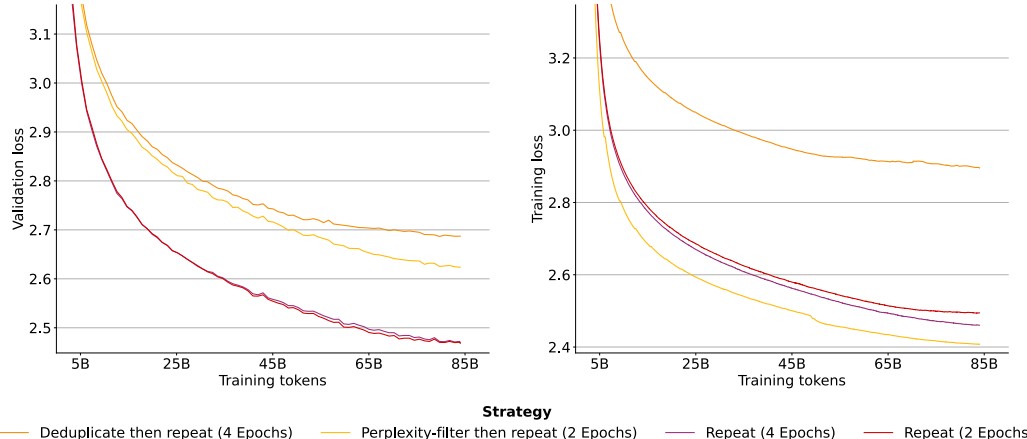

Figure 20: **Validation and training loss of models trained with different data strategies.** Training loss is smoothed with exponential moving average smoothing and a weight of 0.999. Downstream performance of the models is in Figure 6.

To compare complementary data strategies in §7, we have used downstream performance on natural language tasks detailed in Appendix K instead of loss. This is because validation loss gives an unfair advantage to models trained on a larger fraction of data from the same distribution. For example, when making up for missing natural language data with code, models that are trained on more code will have better validation loss on code data while having worse loss on the natural language data as seen in Figure 19: The model pre-trained on 90% of Python code data and 10% of C4 has the highest C4 validation loss, but the lowest Python validation loss.

Models trained on deduplicated or perplexity-filtered data have higher validation loss as the held-out validation data has not gone through the same filtering steps. Thus, its distribution more closely resembles the training data of models trained on the unfiltered data resulting in worse validation loss for the two filtering strategies in Figure 20 (left). Meanwhile, for training loss in Figure 20 (right) the model trained on perplexity-filtered data has the lowest loss. Its training data has been filtered to the top 25% of examples with the lowest perplexity (Appendix N) thus high loss examples have been explicitly filtered out from the training data resulting in low training loss. The model trained on deduplicated data has the highest validation and training loss. This is because commonly

repeated sequences have been filtered out from its training data. Thus, when encountering these common sequences in the unfiltered validation set, its loss is comparatively high as other models have likely simply memorized them. Similarly, fewer repeated sequences during training results in higher training loss as unseen sequences are harder to predict.

## Q   Limitations and Future Work

**Repeating fractions of the data**   In this work we focus on repeating the entire unique dataset for several epochs. Alternatively, one can repeat only a fraction of the dataset. For example, repeating 10% of the dataset for 10 epochs while repeating the rest only for a single epoch as done by Hernandez et al. [40]. To predict loss in that scenario, one may need to adapt our scaling laws with an additional parameter to account for the fraction that is repeated and possibly a parameter that captures at what point in training the data is repeated. Repeating earlier in training when most model weights are still randomly initialized is likely to cause less damage than later in training. Adapting our parametric fit to make concrete scaling predictions for such scenarios is an exciting future research direction.

**Sensitivity to hyperparameters**   The returns from additional epochs may heavily depend on hyperparameters such as learning rate, dropout, or the optimizer choice. It is likely that increasing the learning rate, for example, would lead to diminishing returns from additional epochs kicking in earlier. In this work, we have fixed most hyperparameters to commonly used values for the training of LLMs and leave such explorations to future work.

**Other datasets**   The optimal data strategy is dependent on the dataset at hand and we cannot give universally applicable filtering recommendations. By looking into C4 and OSCAR, we have covered two of the most commonly used English text datasets. Our findings on both datasets were overall in agreement with each other. We have highlighted some of the differences, such as deduplication being more effective on OSCAR due to it being more noisy than C4. Further, we have focused on large-scale pre-training datasets. There is a lot of research on the optimal fine-tuning dataset and methodology for LLMs [94, 62, 128, 85, 117, 68, 116, 134, 115, 36, 125, 72, 63]. More investigations of resolving data-constraints when fine-tuning LLMs may be of interest for future work.

**Other modalities or architectures**   Our work focuses on text datasets and uses the GPT transformer architecture [88]. Prior work has experimented with many variations to the GPT or transformer architecture [27, 104, 96], as well as scaling laws for non-text datasets [1]. Overall, variations of the GPT or transformer architecture have proven very robust and generalizable to other domains [43, 18, 70, 66, 71, 104, 26]. Nonetheless, it may be of interest for future work to test the applicability of our findings in this work to different data modalities or model architectures.

**Other strategies**   There are numerous strategies to solve data constraints not covered in this work that are worth exploring. Like we have shown for Python, future research may consider to what extent augmenting with a natural language (e.g. Chinese) improves performance in another language (e.g. English) and what is the best language to choose [61, 124]. Similarly, while we have looked at deduplication and perplexity filtering, other filtering strategies, such as popularity-based filters [3, 133] and toxicity filters [33, 38, 64, 87, 86] are worth exploring.

## R   Contributions

**Niklas Muennighoff** led experiments, analysis, writing, and the overall project. He implemented, trained and evaluated all models.

**Alexander M. Rush** contributed to framing, results analysis, and paper writing.

**Boaz Barak** contributed to formal and experimental analysis as well as paper writing.

**Teven Le Scao** provided guidance, led data choices and preprocessing, and contributed to framing and writing.

**Aleksandra Piktus** created perplexity and deduplication datasets and contributed to writing.

**Nouamane Tazi** contributed to enabling high-performance training on AMD hardware.

**Sampo Pyysalo** contributed to enabling high-performance training and early repetition experiments.

**Thomas Wolf** provided guidance on experimental design and contributed to paper writing.

**Colin Raffel** provided guidance on experimental design and contributed to paper writing.

## S   Hyperparameters and Setup

For all training runs we use 1% of tokens for linear warm-up of the learning rate to a maximum learning rate of 2e-4 that is decayed to 2e-5 following a cosine schedule. We use a batch size of 256 for models with fewer than 2 billion parameters, 512 for models with 2 - 5 billion parameters and 1024 for models with more than 5 billion parameters. All models are trained in bfloat16 precision using the Adam optimizer [48] with $eps = 1e - 8$, $beta1 = 0.9$. For $beta2$, we found a value of 0.95 to result in slightly lower final loss and fewer loss spikes than the default value of 0.999 in implementations such as PyTorch. However, except for models with FLOP budgets of $C = 9.3 \times 10^{20}$ and $2.1 \times 10^{21}$, we always use $beta2 = 0.999$. We use a dropout rate of 0.1, a weight decay rate of 0.1 and clip gradients at 1.0. These hyperparameter choices are largely based on prior work [42, 110] and performance on test runs. As none of our hyperparameter choices is particularly exotic, we expect our setup to generalize to many other setups. In Table 15 we list the model architectures we use. They are an extended version of the architectures from [42]. We calculate model parameters following [78], which includes embedding parameters:

$$P = 12lh^2 \left( 1 + \frac{13}{12h} + \frac{V + s}{12lh} \right) \tag{23}$$

where $P$ is the final parameter count, $l$ are layers, $h$ is the hidden dimension, $V = 50257$ the vocabulary size and $s = 2048$ the sequence length. We find the parameter counts reported in Chinchilla [42] to be significantly different than our calculations, especially at larger scales. We report both in Table 15, but we use our parameter estimates everywhere in this work. Further, we have corrected the number of heads of the 3,530 and 4,084 million parameter models from [42] to obey the relationship $d\_model = kv\_size \cdot n\_heads$.

To train our models, we have forked the Megatron-DeepSpeed [91, 99] framework and adapted it for ROCm to enable training on AMD GPUs. We have made our training code publicly available at `https://github.com/TurkuNLP/Megatron-DeepSpeed`. Models are trained using data, tensor and pipeline parallelism on up to 256 AMD Instinct MI250X GPUs distributed across up to 64 nodes on the LUMI supercomputer located in Finland. As of June 2023, LUMI is the largest supercomputer in Europe and ranks third worldwide with a performance of around 310 PFLOPs.[5] We trained models in parallel using up to 2,200 nodes at a single point in time (equivalent to around 8,800 GPUs or 17,600 GCDs or 86% of all GPUs on LUMI). We have used a total of around 3 million GPU hours. The cluster is powered 100% by renewable energy (hydroelectricity) and its waste heat is used for heating the nearby city reducing the city's carbon emissions by up to 20%. Thanks to the low temperatures in Finland, relatively little cooling for the cluster is required further reducing its impact on the environment. As of June 2023, it ranks as the seventh greenest supercomputer.[6]

---

[5]https://www.top500.org/lists/top500/2023/06/
[6]https://www.top500.org/lists/green500/2023/06/

| Parameters (millions) | | d_model | ffw_size | kv_size | n_heads | n_layers |
|---|---|---|---|---|---|---|
| This work | Chinchilla | | | | | |
| 7 | - | 128 | 512 | 32 | 4 | 3 |
| 14 | - | 224 | 896 | 32 | 7 | 4 |
| 20 | - | 288 | 1152 | 32 | 7 | 5 |
| 38 | - | 448 | 1792 | 32 | 7 | 6 |
| 52 | 44 | 512 | 2048 | 64 | 8 | 8 |
| 66 | 57 | 576 | 2304 | 64 | 9 | 9 |
| 83 | 74 | 640 | 2560 | 64 | 10 | 10 |
| 97 | 90 | 640 | 2560 | 64 | 10 | 13 |
| 112 | 106 | 640 | 2560 | 64 | 10 | 16 |
| 125 | 117 | 768 | 3072 | 64 | 12 | 12 |
| 146 | 140 | 768 | 3072 | 64 | 12 | 15 |
| 168 | 163 | 768 | 3072 | 64 | 12 | 18 |
| 182 | 175 | 896 | 3584 | 64 | 14 | 14 |
| 201 | 196 | 896 | 3584 | 64 | 14 | 16 |
| 220 | 217 | 896 | 3584 | 64 | 14 | 18 |
| 255 | 251 | 1024 | 4096 | 64 | 16 | 16 |
| 280 | 278 | 1024 | 4096 | 64 | 16 | 18 |
| 305 | 306 | 1024 | 4096 | 64 | 16 | 20 |
| 421 | 425 | 1280 | 5120 | 128 | 10 | 18 |
| 480 | 489 | 1280 | 5120 | 128 | 10 | 21 |
| 502 | 509 | 1408 | 5632 | 128 | 11 | 18 |
| 539 | 552 | 1280 | 5120 | 128 | 10 | 24 |
| 574 | 587 | 1408 | 5632 | 128 | 11 | 21 |
| 619 | 632 | 1536 | 6144 | 128 | 12 | 19 |
| 645 | 664 | 1408 | 5632 | 128 | 11 | 24 |
| 704 | 724 | 1536 | 6144 | 128 | 12 | 22 |
| 789 | 816 | 1536 | 6144 | 128 | 12 | 25 |
| 865 | 893 | 1792 | 7168 | 128 | 14 | 20 |
| 981 | 1018 | 1792 | 7168 | 128 | 14 | 23 |
| 1096 | 1143 | 1792 | 7168 | 128 | 14 | 26 |
| 1215 | 1266 | 2048 | 8192 | 128 | 16 | 22 |
| 1364 | 1424 | 2176 | 8704 | 128 | 17 | 22 |
| 1366 | 1429 | 2048 | 8192 | 128 | 16 | 25 |
| 1517 | 1593 | 2048 | 8192 | 128 | 16 | 28 |
| 1535 | 1609 | 2176 | 8704 | 128 | 17 | 25 |
| 1650 | 1731 | 2304 | 9216 | 128 | 18 | 24 |
| 1706 | 1794 | 2176 | 8704 | 128 | 17 | 28 |
| 1905 | 2007 | 2304 | 9216 | 128 | 18 | 28 |
| 2160 | 2283 | 2304 | 9216 | 128 | 18 | 32 |
| 2179 | 2298 | 2560 | 10240 | 128 | 20 | 26 |
| 2494 | 2639 | 2560 | 10240 | 128 | 20 | 30 |
| 2809 | 2980 | 2560 | 10240 | 128 | 20 | 34 |
| 3090 | - | 2688 | 10752 | 128 | 22 | 34 |
| 3263 | 3530 | 2688 | 10752 | 128 | 21 | 36 |
| 3574 | 3802 | 2816 | 11264 | 128 | 22 | 36 |
| 3900 | 4084 | 2944 | 11776 | 128 | 23 | 36 |
| 4239 | 4516 | 3072 | 12288 | 128 | 24 | 36 |
| 6355 | 6796 | 3584 | 14336 | 128 | 28 | 40 |
| 8672 | 9293 | 4096 | 16384 | 128 | 32 | 42 |
| 10912 | 11452 | 4352 | 17408 | 128 | 32 | 47 |
| 11455 | 12295 | 4608 | 18432 | 128 | 36 | 44 |
| 12220 | 12569 | 4608 | 18432 | 128 | 32 | 47 |
| 13601 | 13735 | 4864 | 19456 | 128 | 32 | 47 |
| 14917 | 14940 | 4992 | 19968 | 128 | 32 | 49 |
| 15056 | 16183 | 5120 | 20480 | 128 | 40 | 47 |

Table 15: **Model architectures.** We list the architectures of all models trained as part of this work. Many shown models have been trained multiple times on different amounts of unique data and for varying epochs.

# T   Prompts and Samples

The following figures illustrate the prompts with samples from each evaluation dataset. Prompts stem from PromptSource [5] or GPT-3 [15]. All data comes from the ground truth datasets in this section, and no generations are shown here.

| | |
|---|---|
| Context → | Edmond (or Edmund) Halley, FRS (pronounced ; 8 November [O.S. 29 October] 1656 - 25 January 1742 [O.S. 14 January 1741] ) was an English astronomer, geophysicist, mathematician, meteorologist, and physicist who is best known for computing the orbit of Halley's Comet.  He was the second Astronomer Royal in Britain, succeeding John Flamsteed. Question:  Edmond Halley was born outside of the United Kingdom.  True, False, or Neither? Answer: |
| Correct Answer → | Neither |
| Incorrect Answer → | True |
| Incorrect Answer → | False |

Figure 21: Formatted dataset example from ANLI R1 evaluated using accuracy as described in Appendix K.

| | |
|---|---|
| Context → | The 1970 Swedish Open was a combined men's and women's tennis tournament played on outdoor clay courts held in Båstad, Sweden and was part of the Grand Prix circuit of the 1970 Tour.  It was the 23rd edition of the tournament and was held from 2 July through 12 July 1970.  Dick Crealy and Peaches Bartkowicz won the singles titles. Question:  Dick Crealy and Peaches Bartkowicz beat eachother in the 1970 Swedish Open.  True, False, or Neither? Answer: |
| Correct Answer → | False |
| Incorrect Answer → | True |
| Incorrect Answer → | Neither |

Figure 22: Formatted dataset example from ANLI R2 evaluated using accuracy as described in Appendix K.

| | |
|---|---|
| Context → | Tokyo - Food group Nestle is seeking to lure Japanese holiday shoppers with a taste for fine snacking with a gold-wrapped Kit Kat chocolate bar.  The single finger Kit Kat is wrapped in a thin layer of gold leaf.  Only 500 of the bars go on sale from Dec.  29 with a price tag of around 2,016 yen ($16).  The Kit Kat chocolate bar made its debut in Japan in 1973 and since then a variety of flavors - from green tea to wasabi - have been produced. Question:  Japanese like kit kat.  True, False, or Neither? Answer: |
| Correct Answer → | True |
| Incorrect Answer → | False |
| Incorrect Answer → | Neither |

Figure 23: Formatted dataset example from ANLI R3 evaluated using accuracy as described in Appendix K.

| | |
|---|---|
| Context → | An astronomer observes that a planet rotates faster after a meteorite impact. Which is the most likely effect of this increase in rotation? |
| Correct Answer → | Planetary days will become shorter. |
| Incorrect Answer → | Planetary years will become longer. |
| Incorrect Answer → | Planetary gravity will become stronger. |

Figure 24: Formatted dataset example from ARC-Challenge evaluated using accuracy as described in Appendix K.

| | |
|---|---|
| Context → | To express the distance between the Milky Way galaxy and other galaxies, the most appropriate unit of measurement is the |
| Correct Answer → | light-year. |
| Incorrect Answer → | meter. |
| Incorrect Answer → | kilometer. |
| Incorrect Answer → | astronomical unit. |

Figure 25: Formatted dataset example from ARC-Easy evaluated using accuracy as described in Appendix K.

| | |
|---|---|
| Context → | Radio wave - Radio waves are a type of electromagnetic radiation with wavelengths in the electromagnetic spectrum longer than infrared light. Radio waves have frequencies as high as 300 gigahertz (GHz) to as low as 30 hertz (Hz). At 300 GHz, the corresponding wavelength is 1 mm, and at 30 Hz is 10,000 km. Like all other electromagnetic waves, radio waves travel at the speed of light. They are generated by electric charges undergoing acceleration, such as time varying electric currents. Naturally occurring radio waves are emitted by lightning and astronomical objects. Question: do radio waves travel at the speed of light? Answer: |
| Correct Answer → | yes |
| Incorrect Answer → | no |

Figure 26: Formatted dataset example from BoolQ evaluated using accuracy as described in Appendix K.

| | |
|---|---|
| Context → | A: Okay. So Frank, what, uh, type of, uh, budget do you or your family have? B: Well, uh I don't know that we really have a budget. Question: he and his family really have a budget. True, False or Neither? Answer: |
| Correct Answer → | False |
| Incorrect Answer → | True |
| Incorrect Answer → | Neither |

Figure 27: Formatted dataset example from CB evaluated using accuracy as described in Appendix K.

| | |
|---|---|
| Context → | The computer was expensive to fix therefore |
| Correct Answer → | I bought a new one. |
| Incorrect Answer → | I got it repaired. |

Figure 28: Formatted dataset example from COPA evaluated using accuracy as described in Appendix K.

| | |
|---|---|
| Context → | Canoeing:  Two women in a child are shown in a canoe while a man pulls the canoe while standing in the water, with other individuals visible in the background.  The child and a different man |
| Correct Answer → | sit in a canoe while the man paddles. |
| Incorrect Answer → | are then shown paddling down a river in a boat while a woman talks. |
| Incorrect Answer → | are driving the canoe, they go down the river flowing side to side. |
| Incorrect Answer → | walking go down the rapids, while the man in his helicopter almost falls and goes out of canoehood. |

Figure 29: Formatted dataset example from HellaSwag evaluated using accuracy as described in Appendix K.

| | |
|---|---|
| Context → | Question:  How to sleep in proper posture? Answer: |
| Correct Answer → | Sleep straight with a pillow under your head. |
| Incorrect Answer → | Sleep straight with a pillow over your head. |

Figure 30: Formatted dataset example from PiQA evaluated using accuracy as described in Appendix K.

| | |
|---|---|
| Context → | As spacecraft commander for Apollo XI, the first manned lunar landing mission, Armstrong was the first man to walk on the Moon.  "That's one small step for a man, one giant leap for mankind." With these historic words, man's dream of the ages was fulfilled. Question:  Neil Armstrong was the first man who landed on the Moon.  True or False? Answer: |
| Correct Answer → | True. |
| Incorrect Answer → | False. |

Figure 31: Formatted dataset example from RTE evaluated using accuracy as described in Appendix K.

| | |
|---|---|
| Context → | The electromagnetic spectrum encompasses a very wide range of wavelengths and frequencies.  Visible light is only a very small portion of the spectrum with wavelengths from 400-700 nm. Question:  With wavelengths from 400-700 nm, what kind of light represents only a very small portion of the spectrum? Answer: |
| Correct Answer → | visible light. |
| Incorrect Answer → | ultraviolet light. |
| Incorrect Answer → | invisible light. |
| Incorrect Answer → | sunlight. |

Figure 32: Formatted dataset example from SciQ evaluated using accuracy as described in Appendix K.

| | |
|---:|:---|
| Context → | Bob went to the gas station to fill up his car. His tank was completely empty and so was his wallet. The cashier offered to pay for his gas if he came back later to pay. Bob felt grateful as he drove home. Answer: |
| Correct Answer → | Bob believed that there were good people in the world. |
| Incorrect Answer → | Bob contemplated how unfriendly the world was. |

Figure 33: Formatted dataset example from StoryCloze evaluated using accuracy as described in Appendix K.

| | |
|---:|:---|
| Correct Context → | Johnny likes fruits more than vegetables in his new keto diet because the fruits: |
| Incorrect Context → | Johnny likes fruits more than vegetables in his new keto diet because the vegetables: |
| Target Completion → | are saccharine. |

Figure 34: Formatted dataset example from WinoGrande evaluated using accuracy as described in Appendix K.

| | |
|---:|:---|
| Context → | Given the following data about a restaurant: name : The Wrestlers eatType : pub food : Japanese priceRange : cheap area : riverside near : Raja Indian Cuisine

Generate some text about this restaurant. |
| Target → | The Wrestlers offers Japanese food and pub with cheap price near Raja Indian Cuisine in riverside. |

Figure 35: Formatted dataset example from E2E NLG evaluated using ROUGE as described in Appendix K.

```
Context →    Article:  The artificial intelligence system - LipNet - watches video
             of a person speaking and matches the text to the movement of their
             mouths with 93% accuracy, the researchers said.
             Automating the process could help millions, they suggested.
             But experts said the system needed to be tested in real-life
             situations.
             Lip-reading is a notoriously tricky business with professionals only
             able to decipher what someone is saying up to 60% of the time.
             "Machine lip-readers have enormous potential, with applications in
             improved hearing aids, silent dictation in public spaces, covert
             conversations, speech recognition in noisy environments, biometric
             identification and silent-movie processing," wrote the researchers.
             They said that the AI system was provided with whole sentences so
             that it could teach itself which letter corresponded to which lip
             movement.
             To train the AI, the team - from Oxford University's AI lab - fed it
             nearly 29,000 videos, labelled with the correct text.  Each video was
             three seconds long and followed a similar grammatical pattern.
             While human testers given similar videos had an error rate of 47.7%,
             the AI had one of just 6.6%.
             The fact that the AI learned from specialist training videos led some
             on Twitter to criticise the research.
             Writing in OpenReview, Neil Lawrence pointed out that the videos had
             "limited vocabulary and a single syntax grammar".
             "While it's promising to perform well on this data, it's not really
             groundbreaking.  While the model may be able to read my lips better
             than a human, it can only do so when I say a meaningless list of
             words from a highly constrained vocabulary in a specific order," he
             writes.
             The project was partially funded by Google's artificial intelligence
             firm DeepMind.

             Summary:

Target →     Scientists at Oxford University have developed a machine that can
             lip-read better than humans.
```

Figure 36: Formatted dataset example from XSUM evaluated using ROUGE as described in Appendix K.

```
Context →    I will verbalize an abstract representation of a sentence in natural
             language.  To do so, I will first show the representation and
             then the natural language.  The text needs to include all of the
             information in the representation.
             Brandon_Carter | almaMater | University_of_Cambridge,
             University_of_Cambridge | chancellor | David_Sainsbury,_Baron_Sainsbury_of_Turville,
             Brandon_Carter | birthPlace | England, University_of_Cambridge |
             viceChancellor | Leszek_Borysiewicz

Target →     The University of Cambridge is the alma mater of Brandon Carter,
             who was born in England.  David Sainsbury, also known as the Baron
             Sainsbury of Turville, and Leszek Borysiewicz are respectively the
             chancellor and vice chancellor of the University of Cambridge.
```

Figure 37: Formatted dataset example from WebNLG evaluated using ROUGE as described in Appendix K.

| | |
|---|---|
| Context → | Attributes are placed within the tag itself, making additional alterations to the ëlement contenẗbetween the start and end tag. They never stand alone.  They are written in the format name=v̈alue,̈ where name is the name of the attribute (for instance c̈olor)̈, and value describes this specific instance (for instance r̈ed)̈.  You've actually seen attributes before, if you followed the tutorial in the basic HTML section.   tags use the src attribute, anchors use the name attribute, and links use the href attribute.  See how those all follow the \_\_\_=␣\_\_f̈ormat?  Making a table, or chart, requires several different tags.  Play with these tags, or learn about HTML tables in more detail.  Start with table tags around the entire table:<table></table> Row tags around the contents of each row:  <tr> Column headers in the first row:  <th> Cells in subsequent rows:  <td> Here's an example of how it all fits together:<table><tr><th>Column 1:  Month</th><th>Column 2:  Money Saved</th></tr><tr><td>January</td><td>$100</td></tr></table> You've already learned the <head> tag, which shows up at the start of each document.  Besides the <title> tag, it can include the following types of tags:  Meta tags, which are used to provide metadata about a web page.  This data can be used by search engines when the robot scours the internet to locate and list websites.  To make your website more visible on search engines, use one or more <meta> start tags (no end tags necessary), each with exactly one name attribute and one content attribute, for example:  <meta name=d̈escriptionc̈ontent=ẅrite a description here;̈ or <meta name=k̈eywordsc̈ontent=ẅrite a list of keywords, each separated by a comma>̈ <link> tags are used to associate other files with the page.  This is mainly used to link to CSS stylesheets, which are made using a different type of coding to alter your HTML page by adding color, aligning your text, and many other things.  <script> tags are used to link the page to JavaScript files, which can cause the page to change as...

TL;DR in English: |
| Target → |  Learn about attributes.  Experiment with HTML tables.  Learn the miscellaneous head tags.  Play around with HTML found on websites.  Learn more advanced web design from comprehensive guides. |

Figure 38: Formatted dataset example from WikiLingua evaluated using ROUGE as described in Appendix K.

| | |
|---|---|
| Context → | John travelled to the kitchen.  Sandra moved to the kitchen.  Daniel went to the kitchen.  John journeyed to the hallway.  Mary journeyed to the bedroom.  Mary journeyed to the kitchen.  Mary travelled to the bedroom.  Sandra travelled to the bedroom.  John went to the office.  John went back to the kitchen.  Where is Mary? |
| Target → |  bedroom |

Figure 39: Formatted dataset example from bAbI evaluated using exact match as described in Appendix K.

## U Other Experiments

**UL2**   We experimented with the UL2 objective [106, 107] for a causal model but did not find it to outperform regular causal language modeling on our evaluation tasks. This may stem from UL2 being better suited as an Encoder-Decoder model or from mistakes in our UL2 implementation.

**The Pile**   We have also trained several models on The Pile [31] and found similar trends as for OSCAR and C4. We make these models publicly available.

## V Release of Artifacts

We open-source all of our models and code under Apache 2.0 licenses. Our filtered datasets are released with the same licenses as the datasets they stem from. All material can be found at: `https://github.com/huggingface/datablations`.

## W Version Control

**V3 → V4:**

- Added comparison of different fits in terms of loss and $R^2$ in Table 1
- Small writing improvements

**V2 → V3:**

- Added loss curves of complementary strategies in Appendix P
- Fixed OSCAR validation plot in Appendix I
- Clarified the usage of smoothing in training and validation plots in Appendix K
- Added more references

**V1 → V2:**

- Added experiments decaying alpha and beta to allow excess epochs or paramters to hurt in Appendix F
- Added Galactica case study in Appendix G
- Added more details on the calculation of $U_N$ given $U_D$ in Appendix A
- Added hyperparameter sensitivity limitation in Appendix Q
- Added more detail on how score normalization is done in Appendix K
- Mentioned modification of number of heads in Appendix S

## X Broader Impacts

Large Language Models carry potential risks such as outputting offensive language, propagating social biases, and leaking private information [119, 8]. By publicly releasing all of our models and providing new insights to improve the scaling of LLMs we may contribute to the further proliferation of these harms. However, we note that there are already much larger and more capable models freely available [14, 13, 95, 11] that can be used in such harmful ways. Thus, we consider the open-source release of our models and research to significantly outweigh its downsides.