# OpenReview forum: "Scaling Data-Constrained Language Models"
_NeurIPS.cc/2023/Conference — NeurIPS 2023 oral_

### Official Review · Reviewer_r7xA · 2023-06-13

**Soundness:** 4 excellent
**Presentation:** 4 excellent
**Contribution:** 4 excellent
**Rating:** 7
**Confidence:** 4

**Summary:**

This paper studies an important empirical question: what is the optimal tradeoff between LLM size and the amount of data available for training it. The authors follow an existing line of work (e.g., Chinchilla), and additionally consider the role of data repetition (via multiple epochs), e.g., what is the effect of training a model for K epochs on N tokens each, compared to training it for one epoch on K*N unique tokens. The authors adjust the power law equation introduced in Chinchilla to take into account the “effective data”, which considers the lower value of repeated tokens, and similarly the “effective model size”, which considers the lower value of parameters trained on such data. They present an overwhelmingly large set of experiments, some VERY expensive (the total amount of compute spent on this work is 4.5 million GPU hours). This makes this study practically irreproducible for 99.999% of the community on the one hand, but very valuable still to those training such huge models. The conclusions reached by the authors are interesting: e.g., models can be trained on data that is repeated for up to 4 times, and reach a similar loss as models trained on a similar amount of unique tokens. They also study the effect of augmenting text corpora with code, and applying deduplication filtering steps.


**Strengths:**

- An important empirical question.
- A potentially more accurate modeling of the params/data scaling law.
- A very large set of experiments covering a range of models and scenarios.
- Some interesting and non-trivial observations.
- Paper is generally well written



**Weaknesses:**

- There is something unclear about using the validation or the test loss. The authors say (#42) that they are reporting test loss, but many of the tables and graphs say that they are showing validation loss. Which one is it? Reporting test loss is a problem, as it might lead to overfitting the test data (e.g., by others who want to adapt the proposed recipes).
- The loss differences are often very small. E.g., in Fig1 (right), a difference of 0.006, or in Fig5 (left). Are these results significant?


**Questions:**

- I am not sure I entirely understand the data repetition setup. In particular, the caption of fig2 says “Training runs with different epochs reuse subsets of the same data to ensure different training data is not a confounding factor.” What does subsets of the same data mean? Is the data sampled from the training data in each repetition? Doesn’t that mean that the data is not effectively repeated?

Typos and such:
- #132-133: I think R_D and U_D in these lines should be R_N and U_N?
- #147: [65]partir

---

> ### Author Rebuttal · Authors · 2023-08-07
>
> Thank you for your thorough review.
>
> Weaknesses:
>
> > There is something unclear about using the validation or the test loss. The authors say (#42) that they are reporting test loss, but many of the tables and graphs say that they are showing validation loss. Which one is it? Reporting test loss is a problem, as it might lead to overfitting the test data (e.g., by others who want to adapt the proposed recipes).
>
> We are sorry for the lack of clarity. We have rephrased the mention of test loss to “we report loss on a held-out test set unless otherwise specified”. In the main text, only Figure 4 reports validation loss throughout training instead. We have also added a detailed explanation of how loss is computed in a new section Appendix K “Evaluation Details”. We are not sure why reporting the test loss would be a problem. We are using a fully held-out dataset and compute the loss over it for each model at the end of training. Apart from loss, we also show that our findings are mirrored in downstream performance in the Appendix. Given that all our models will be open-sourced, future work can also run them and compute loss on their own datasets if desired.
>
>
> > The loss differences are often very small. E.g., in Fig1 (right), a difference of 0.006, or in Fig5 (left). Are these results significant?
>
> We have updated Fig1 (right) to larger models with 8.67B and 6.34B parameters. The loss difference for that setup is 0.017. At these scales, loss decreases very slowly, for example for the 8.67B model, its last 5,000 steps ($\approx$10B tokens) correspond to about a 0.017 loss decrease. Thus, we consider it to be significant. We have also added downstream performance of these two models in the Appendix. Across our 19 downstream evaluation tasks, the 6.34B model has an average score of 25.9, while the 8.7B model scores 23.5. This reinforces our findings that data should be scaled faster than parameters in the repeated regime.
>
> Questions:
>
> We are sorry for the confusion. We meant to say that if we have e.g. 10B unique tokens, those same 10B unique tokens are also a subset of every setup where we have >10B unique tokens. This ensures that as much as possible of the data is shared across runs and our results are not impacted by simply having trained on a better data subset. We have rephrased the caption of that Figure as “We ensure that runs using less data (more epochs) always use a subset of the data used in runs with more data (fewer epochs).”.
>
>
> Typos:
> - #132-133: The behavior for R_D = R_D* is the same as the behavior for R_N = R_N* due to the symmetry hence either one is fine in these lines.
> - #147: Thank you for noting, we have fixed it.

---

> > ### Comment · Reviewer_r7xA · 2023-08-19
> >
> > Thank you for the clarifications, I am happy with the responses.

---

### Official Review · Reviewer_v1HT · 2023-07-06

**Soundness:** 4 excellent
**Presentation:** 4 excellent
**Contribution:** 4 excellent
**Rating:** 8
**Confidence:** 3

**Summary:**

This paper demonstrates the scaling law both mathematically and experimentally in data-constrained situations. They analyze three experimental protocols: allocation (fixed unique data), return (fixed FLOPs), and parametric fit, and obtain new conclusions that were not found in previous research. For example, they demonstrate that increasing the data repetition (epochs) is more effective than increasing the parameters when the number of unique tokens is fixed, and the performance drop is shown to be negligible when there is a slight decrease in unique data within a predetermined number of FLOPs. They also analyze the effects of code data augmentation and data filtering strategies that can be applied in data-constrained situations.

**Strengths:**

- The research is well-motivated and necessary.
- They present scaling laws and propose optimal training strategies in a situation where data is limited.
- They derive reliable and generalizable conclusions through extensive experiments.

**Weaknesses:**

- Increasing the maximum parameter size (e.g. >= 10B in Chinchilla [1]) would further enhance comprehension of the scaling law in large language models.

[1] Hoffmann, Jordan, et al. "An empirical analysis of compute-optimal large language model training." Advances in Neural Information Processing Systems 35 (2022): 30016-30030.

**Questions:**

Since many large language models utilize IF (instruction fine-tuning) and RLHF (reinforcement learning from human feedback), do authors have plans to expand research on scaling laws of data-constrained language models in the IF or RLHF scenario, where the data includes instruction or human feedback data?

**Limitations:**

The authors adequately suggest limitations in various aspects.

---

> ### Author Rebuttal · Authors · 2023-08-07
>
> Thank you for your review. We agree that future work confirming these findings at >10B would be very interesting. The Galactica 120B model from prior work does provide preliminary evidence that you can repeat at these larger scales, but more experiments are needed.
>
> Questions:
>
> >  Data-Constrained IF (instruction fine-tuning) and RLHF
>
> * While we are very interested in IF and RLHF, we think the scaling environment for these problems is different enough to warrant an analysis outside of the Chinchilla regime. For IF, often the datasets are quite small and training looks more like traditional ML training (for example the LIMA dataset from Meta only contains 1000 examples). For RLHF, the reward model is "data-constrained" in the sense that annotations are costly. However this model looks more like a classification/regression model, and would likely have different scaling properties than next-token prediction models. Furthermore, published models only scale up to the order of 1M examples which is a different order of magnitude than these examples. Scaling the RL phase of the system is quite interesting, however, this phase also has a different objective and training procedure (PPO).

---

> > ### Comment · Reviewer_v1HT · 2023-08-20
> >
> > Thank you for updates and answer about the question. I raised my score after reading the rebuttal.

---

### Official Review · Reviewer_ATmT · 2023-07-06

**Soundness:** 3 good
**Presentation:** 3 good
**Contribution:** 4 excellent
**Rating:** 8
**Confidence:** 4

**Summary:**

This paper investigates scaling properties of languages, in the presence of *repeated data*: plenty of work has studied the scaling properties of LLMs (and ML architectures in general), both with respect to the number of parameters and amount of data, but they generally assume each new datapoint is unique. However, as newer LLMs start to reach the limit of available (internet) data, repeated use of the same datapoints sets will become more common and understanding the "worth" of a repeated datapoint is a very important research question.

The authors conduct extensive experiments, training hundreds of model ranging from 10 million to 9 billion parameters and trained up for up to 1500 (repeated) epochs. They then propose a novel *data-constrained* scaling law by introducing a multiplicative factor the data term that decays with the number of repetitions and (briefly) validate it, showing a better fit than the typical Chinchilla scaling law in the presence of repeated data.

They then use their proposed scaling to study two questions:

1. *Allocation: What is the optimal balance of (compute/data) resources?*
Their results suggest that its better to first scale epochs/number of repetions, and only then parameter count: by repeating data, they are able to train a comparable model (in terms of performance and compute budge) with 27% less parameters.
2. *Return: What is the expected value of additional resources?*
They discover that while there are diminishing returns to adding more compute resources in the presence of the same training dataset, there's a notable value in repeating data: training up to 4 epochs yields almost the same loss as training on unique data, but beyond that the value of adding compute resources to train further effectively drops to zero.

Lastly, they explore several methods to address data limitations without the need to produce new natural language data. They found that incorporating code tokens into training data yielded a two-fold increase in effective tokens, even for tasks solely evaluating natural language. They also explored varying data filtering techniques, concluding that data filtering primarily yields substantial benefits on noisy datasets, and not so much on cleaner ones.

**Strengths:**

This was very interesting read, and is quite well written. I think this work has the potential to have big impact in the LLM pretraining community: the limits of unique data we can get from the internet are already being reached, but the findings in this paper suggest that won’t necessarily be a problem for a while, and that repeating tokens a few times to obtain an optimal capacity-data compute allocation is probably fine. The experiments on filtering even hint that it might be better to even be more aggressive in current filtering and repeating the cleaner tokens. Their evaluation on Section 7 is also quite strong, using downstream performance, and the findings with code are insighful and also super relevant to the pretraining community, and further confirm “implicit” knowledge in the community.

**Weaknesses:**

The only slightly more serious flaw I see is that the proposed scaling law is not well validated: from what I see, in the main paper only a single plot showing the fit is shown, and no comparable fitting metrics (like r2), commonly reported in scaling law papers, are given to quantitatively compare to the unique-data scaling laws.

I think the paper could also be improved by making the theoretical discussion more concise and moving some of the (very interesting) findings from the appendix to the main text.

**Questions:**

N/A

**Limitations:**

See above

---

> ### Author Rebuttal · Authors · 2023-08-07
>
> Thank you for your thorough summary of the work and careful review.
>
> > by repeating data, they are able to train a comparable model (in terms of performance and compute budget) with 27% less parameters.
>
> We have updated the main plot (right) to larger scales comparing a 8.7 billion parameter model and a 6.3 billion parameter one. The difference is even more significant now with the smaller model performing better. We further show that the difference between these two models is also significant in terms of downstream performance. Thus, we would go beyond calling them comparable but rather that the smaller model is indeed better.
>
>
> > The experiments on filtering even hint that it might be better to even be more aggressive in current filtering and repeating the cleaner tokens.
>
> This is a great observation. Indeed the models that repeat C4 twice are better than the models trained on OSCAR (Appendix) with just one repeat.
>
> **Weaknesses:**
>
> > ..fitting metrics..
>
> We have added R^2 and loss for the different versions of our exponential decay formulation as well as applying no decay at the end of Appendix A. Thank you for this suggestion.
>
>
> > I think the paper could also be improved by making the theoretical discussion more concise and moving some of the (very interesting) findings from the appendix to the main text.
>
> We agree with the sentiment that much of the Appendix could be part of the main text, however, we are unsure how to reduce the theoretical discussion to make space for that without sacrificing content necessary to understand the main text. If you have specific suggestions, we are open to revising the text based on them.

---

> > ### Comment · Reviewer_ATmT · 2023-08-18
> > **Response**
> >
> > > If you have specific suggestions, we are open to revising the text based on them.
> >
> > I guess my (lazy) suggestion is just arxiv'ing a longer (8+) pages :)

---

### Official Review · Reviewer_sNww · 2023-07-07

**Soundness:** 4 excellent
**Presentation:** 3 good
**Contribution:** 4 excellent
**Rating:** 7
**Confidence:** 4

**Summary:**

This paper studies the scaling behavior of language models by repeating the training data to multiple epochs. The authors extend the recent successful Chinchilla scaling law to a data-constrained regime by adding exponential decay of data and model terms then fit the empirical observations. The key takeaway is that repeating data up to 4 times yields negligible changes to loss compared to having unique data, consistent with the observation of Galactica. The authors further studies additional methods like adding code data to 50% and find out this does not compromise model performance on text tasks.

**Strengths:**

This paper studies a very important problem (what should one do to further scale up data when data is used up) and offers very practical advice (repeating the data 4 times and using code 50% to the training data). I believe this paper will make a clear impact in the area of language models and general AI.

**Weaknesses:**

Although this paper has clear advantages in terms of practical guidance, I am concerned about the following weakness:

The choice of the specific parametric form in equation 5 and 6 are not clearly explained and may be subject to debate, specifically:

- Why the decay in equation 5 should be exponential (other than, say, polynomial) as when data points are small, exponential may look similar to polynomial (or other functional forms).
- Why the decay in model scale (Eq. 6) should follow the same form as data (Eq. 5). For the original Chinchilla scaling law, the two terms taking symmetric forms are understandable due to their nature. But for repeating data, model parameter and data scale are not as symmetric as unique data setting, thus considering the same form of decay as data (though I do believe there should be a certain form of decay in model scale) might be less justified

Different types of data may need different levels of repetition.

- The Chinchilla model mostly considers webpages v.s. the Galactica model mostly consider academic papers. Intuitively, the level of complexity of webpages might be lower than papers, and one may want to repeat the complex data over simple ones.

The conclusions hold for smaller models that are less than 10B may not hold for models that are larger than 65B

- This is actually my largest concern with regard to this paper. Empirically, at least for fine-tuning, people have observed that larger models do not require as many fine-tuning epochs as smaller models. In general, models that are smaller than 10B has different behaviors than models larger than 65B
- An alternative explanation for the repeating 4 times conclusion drawn by this paper could be: smaller models do not have enough capability to fit the data with only one pass, but larger models may be able to do so.

The conclusion holds for data scale that is less than 100B may not hold for data scale that is larger than 1T.

- Most of the experiments in this paper consider training models with less than 100B data. In the current practice, models (even if they are just 7B) are mostly trained with 1T data (either code or text). This means that in practice, the problem is whether repeating 1T data 4 times, rather than repeating 10B data 4 times.
- As the data becomes large, a possible model behavior could be: with 10B data, because data is relatively small, repeating is beneficial, yet when data becomes large, repeating may not be as beneficial as the small data regime.

**Questions:**

The authors reported that WebNLG and BaBI immediately improveSee about discussions as one mix in code data. I am curious about what kind of synergy between text and code data and what are the families of text tasks that would benefit the most from mixing code?

**Limitations:**

See above discussion about weakness

---

> ### Author Rebuttal · Authors · 2023-08-07
>
> Thank you for your detailed review and thought-provoking points.
>
> Weaknesses:
>
> Choice of the parametric form:
> - Decay: We agree that these experiments do not rule out other parameteric forms, such as polynomial. We think exponential was an intuitive starting point out of possible decay formulations and the sharp dropoff in data value (which rules out forms like linear decay). We stuck with it given that it fits our data well with few parameters.
> - Parameters: While we do not have a mechanistic explanation for this term, our hypothesis why this decay is necessary is that if you are trying to fit an over-parameterized model to little data then parameters learn redundant features, and so they end up "repeating" in a similar way to the data.
>
> Different types of data:
> - We agree that different types of data is an important factor, although it is challenging to change dataset genre, while controlling for scale. We do run experiment with another large-scale web dataset, OSCAR, which has a different data distribution with less filtering. Despite this difference we found the trends to be about the same. While its difficult to define a general-purpose scalar value of complexity, controlled experiments based on this factor is certainly an interesting research direction for future work. (And personally it would be refreshing if quality had a non-linear scaling factor compared to quantity.)
>
> Larger models:
> - We agree that it is important to further validate these findings at even larger scales than our 8.7B models. We already spent a significant amount of compute for the experiments at hand, so we leave this to future work.
> - We investigated 3 model scales in detail (2.7B, 4.2B, 8.7B) and found very similar behavior. However, it could indeed be that there are fundamental changes when going significantly beyond 10B.
>
> More data:
> - We have swapped the models in the first plot (right) for larger models trained on more data: 8.67 billion parameters for 178 billion tokens and 6.34 billion parameters trained for 242 billion tokens. This has reinforced our findings as we see the same behavior of many epochs still being beneficial (9.7 epochs). We are indeed not hitting the trillion token regimes and it is important for future work (with access to even more compute) to investigate this.
> - This depends on the model size. E.g. with 10B tokens and a 1B model, you will get lots of value from repeating. However, with 1T tokens and a 1B model, then the model will already be close to its maximum capacity at 1T tokens, so repeating is expected to have less value. If we adapt the model size to be bigger in the second case and have more capacity, then our current findings suggest that we can get the same value out of repeating as in the smaller case.
>
>
> Questions:
>
> Our intuition is that code data benefits tasks that require long-term state tracking. Here is an example input from bAbI: “Sandra travelled to the office. Sandra went to the bathroom. Mary went to the bedroom. Daniel moved to the hallway. John went to the garden. John travelled to the office. Daniel journeyed to the bedroom. Daniel travelled to the hallway. John went to the bedroom. John travelled to the office. Where is Daniel?” The correct answer is “hallway". Without code data, the models benchmarked are not able to solve this task. However, as soon as code data is added the ability of state tracking appears to emerge and they can solve such tasks. We believe that this could be due to the necessity of keeping track of variable states in order to accurately predict code.

---

### Author Rebuttal · Authors · 2023-08-07

We thank all reviewers for their thorough reviews. We have made the following updates to the paper:

Increased the scale of the models in the first plot (right) showing even more significant results in support of scaling data faster than parameters when repeating (also in terms of downstream performance shown in a table in the Appendix). As we cannot upload a new version of the paper, we’re attaching the plot and table to this message.

Added several new Appendix sections:
- Appendix G: Case study on how Galactica should have allocated compute according to data-constrained scaling laws
- Appendix K: Details on how loss is computed, how scores are normalized and smoothing for the graphs
- Appendix P: Loss curves of complementary strategies

Improved several Appendix sections:
- Appendix A: More details on the calculation of $U_N$ and comparison of different fits and their loss and $R^2$
- Appendix F: Added experiments on an alternative formulation to let excess parameters hurt performance by decaying alpha and beta
- Appendix I: Fixed one loss plot for OSCAR
- Appendix Q: Added hyperparameter sensitivity as a new limitation
- Appendix R: Specified some more hyperparameters

Rephrased several sentences throughout and fixed typos.

We agree with the overall sentiment expressed by the reviewers that there are many exciting related research questions, such as behavior at hundreds of billions of parameters and trillions of tokens, different datasets beyond C4 and OSCAR and data-constrained scaling laws during finetuning. We look forward to answering these together with the broader research community in future work.

---

### Decision · Program_Chairs · 2023-09-21

**Decision:**

Accept (oral)

**Comment:**

This paper presents a valuable investigation into the scaling of language models in data-constrained regimes, offering insights into the optimal use of compute resources and potential approaches for mitigating data scarcity. The authors conducted a large set of experiments and proposed a scaling law for compute optimality, which was empirically validated. The provision of trained models and datasets used in the experiments for public access adds to the value of this paper. Overall, this work is a very solid contribution to the communnity in pretraining of LLMs.